# RUNX1 marks a luminal castration-resistant lineage established at the onset of prostate development

Renaud Mevel[1], Ivana Steiner[2], Susan Mason[3], Laura CA Galbraith[3], Rahima Patel[1], Muhammad ZH Fadlullah[1], Imran Ahmad[3,4], Hing Y Leung[3,4], Pedro Oliveira[5], Karen Blyth[3,4], Esther Baena[2,6], Georges Lacaud[1]*

[1]Cancer Research United Kingdom, Stem Cell Biology Group, Cancer Research United Kingdom Manchester Institute, The University of Manchester, Alderley Park, Alderley Edge, Macclesfield, United Kingdom; [2]Cancer Research United Kingdom, Prostate Oncobiology Group, Cancer Research United Kingdom Manchester Institute, The University of Manchester, Alderley Park, Alderley Edge, Macclesfield, United Kingdom; [3]Cancer Research United Kingdom Beatson Institute, Bearsden, Glasgow, United Kingdom; [4]Institute of Cancer Sciences, College of Medical, Veterinary and Life Sciences, University of Glasgow, Bearsden, Glasgow, United Kingdom; [5]Department of Pathology, The Christie NHS Foundation Trust, Manchester, United Kingdom; [6]Belfast-Manchester Movember Centre of Excellence, Cancer Research United Kingdom Manchester Institute, The University of Manchester, Alderley Park, United Kingdom

*For correspondence:
georges.lacaud@cruk.manchester.ac.uk

Competing interests: The authors declare that no competing interests exist.

**Abstract** The characterization of prostate epithelial hierarchy and lineage heterogeneity is critical to understand its regenerative properties and malignancies. Here, we report that the transcription factor RUNX1 marks a specific subpopulation of proximal luminal cells (PLCs), enriched in the periurethral region of the developing and adult mouse prostate, and distinct from the previously identified NKX3.1[+] luminal castration-resistant cells. Using scRNA-seq profiling and genetic lineage tracing, we show that RUNX1[+] PLCs are unaffected by androgen deprivation, and do not contribute to the regeneration of the distal luminal compartments. Furthermore, we demonstrate that a transcriptionally similar RUNX1[+] population emerges at the onset of embryonic prostate specification to populate the proximal region of the ducts. Collectively, our results reveal that RUNX1[+] PLCs is an intrinsic castration-resistant and self-sustained lineage that emerges early during prostate development and provide new insights into the lineage relationships of the prostate epithelium.

## Introduction

The prostate is a glandular organ of the mammalian male reproductive system. In mice, prostate development starts during embryogenesis at embryonic day (E) 15.5–16.5 with the emergence of the first prostatic buds from the rostral end of the urogenital sinus (UGS) (*Bhatia-Gaur et al., 1999*; *Georgas et al., 2015*; *Keil et al., 2012*; *Toivanen and Shen, 2017*). These initial buds grow into the surrounding mesenchyme to develop postnatally and through puberty into a branched ductal network organized in distinct pairs of lobes, known as the anterior prostate (AP), dorsolateral prostate (DLP), and ventral prostate (VP) (*Sugimura et al., 1986a*). Each lobe has distinct branching patterns, histopathological characteristics, and is thought to contribute differently to the physiological function of the prostate. The differentiated epithelium of the adult prostate gland is mainly composed of

**eLife digest** The prostate is part of the reproductive organs in male mammals. Many of the cells lining the inside of the prostate – known as 'luminal cells' – need hormones to survive. Certain treatments for prostate cancer, including surgical and chemical castration, lead to fewer hormones reaching the prostate, which shrinks as luminal cells die. But some of these luminal cells are able to survive the damaging effects of castration, rebuilding the prostate upon treatment with hormones, which can lead to the cancer reappearing. It is unclear which type of luminal cells survive during periods without hormones and are responsible for regenerating the prostate.

RUNX1 is a protein responsible for switching genes on and off, and is usually found in blood cells, which it helps to mature and perform their roles, but has also been detected in tissues that depend on hormones. Since the luminal cells of the prostate rely on hormones, could RUNX1 also be present in these cells? To answer this question, Mével et al. used mice to determine where and when RUNX1 is found in prostate cells.

Mével et al. detected high levels of RUNX1 in a patch of luminal cells at the base of the prostate. Samples of these cells were taken for further testing from developing mouse embryos, healthy adult mice and mice in which the prostate was regenerating after surgical castration. Mével et al. found that these cells were a distinct subtype of luminal cells that were able to resist the effects of castration – they survived without hormones. Though these cells were present during the early stages of prostate embryonic development and in healthy adult prostate tissue, they were not responsible for rebuilding the prostate after castration.

Mével et al.'s results indicate that, in mice, RUNX1 may act as a marker for a subset of luminal cells that can survive after castration. Further probing the roles of these castration-resistant luminal cells in normal and cancerous prostate tissue may improve the outcome of patients with prostate cancer treated with hormone deprivation therapy.

basal and luminal cells, interspersed with rare neuroendocrine cells (*Shen and Abate-Shen, 2010*; *Toivanen and Shen, 2017*; *Wang et al., 2001*). Luminal cells form a layer of polarized tall columnar cells that depend on androgen signaling and produce the prostatic secretions. Basal cells act as a supportive layer located between the luminal cells and the surrounding stroma.

Despite being mostly quiescent under homeostatic conditions, the prostate gland encompasses incredible plasticity. In mice, surgical castration-induced prostate involution has proven an invaluable tool to identify progenitor castration-resistant cell populations, characterized by their ability to survive in the absence of androgens, and to fully regenerate an intact adult prostate after re-administration of testosterone (*Barros-Silva et al., 2018*; *Kwon et al., 2016*; *McAuley et al., 2019*; *Tsujimura et al., 2002*; *Wang et al., 2015*; *Wang et al., 2009*; *Yoo et al., 2016*). Such plasticity has also been shown in defined experimental conditions to stimulate regenerative properties of epithelial subpopulations, including transplantations (*Barros-Silva et al., 2018*; *Burger et al., 2005*; *Lawson et al., 2007*; *Lukacs et al., 2010*; *Richardson et al., 2004*; *Wang et al., 2009*; *Xin et al., 2005*; *Yoo et al., 2016*), injury repair (*Centonze et al., 2020*; *Horton et al., 2019*; *Kwon et al., 2014*; *Toivanen et al., 2016*), and organoid assays (*Chua et al., 2014*; *Höfner et al., 2015*; *Karthaus et al., 2014*). In addition, several studies have proposed that progenitor populations with distinct physiological roles and regenerative capacity reside at different locations within the prostate (*Burger et al., 2005*; *Crowell et al., 2019*; *Goldstein et al., 2008*; *Goto et al., 2006*; *Kwon et al., 2016*; *Leong et al., 2008*; *McNeal, 1981*; *Tsujimura et al., 2002*). However, the precise cellular hierarchy and how it is established during development remains controversial.

RUNX transcription factors (TF) are master regulators of lineage commitment and cell fate (*Mevel et al., 2019*). In particular, RUNX1 is essential for the ontogeny of the hematopoietic system and alterations of RUNX1 have been associated with a broad spectrum of hematological malignancies. Interestingly, increasing evidence implicates RUNX1 in the biology and pathology of hormone-associated epithelia (*Lie-A-Ling et al., 2020*; *Riggio and Blyth, 2017*; *Scheitz and Tumbar, 2013*), including breast (*Browne et al., 2015*; *Chimge et al., 2016*; *Ferrari et al., 2014*; *van Bragt et al., 2014*), uterine (*Planagumà et al., 2004*; *Planagumà et al., 2006*), ovarian (*Keita et al., 2013*), and prostate cancers (*Banach-Petrosky et al., 2007*; *Scheitz et al., 2012*; *Takayama et al., 2015*).

Despite the documented importance of RUNX TFs and reports of RUNX1 in PCa, its expression in the normal prostate gland during development, homeostasis, and regeneration has not been explored.

In this study, we found that *Runx1* marks a discrete subset of luminal cells located in the proximal region of the prostatic ducts. Using mouse models, combined with lobe-specific single-cell transcriptomic profiling of adult, castrated, and developing prostates, we show that RUNX1⁺proximal luminal cells represent a distinct lineage established at the onset of prostate development, displaying intrinsic castration-resistant and self-sustaining properties.

## Results

### RUNX1 marks a subpopulation of prostate proximal luminal cells (PLCs)

We initially sought to characterize the expression pattern of *Runx1* in adult mouse prostate. While RUNX1 was detected in basal cells at multiple spatial locations, its expression was specifically high in a subset of luminal cells found in the proximal region of all three lobes, also known as periurethral (*Figure 1A,B*; *Figure 1—figure supplement 1A,B*). Sections were co-stained with NKX3.1, a master regulator of prostate development broadly expressed in luminal cells. Using quantitative image-based cytometry (QBIC), we found that RUNX1 and NKX3.1 had a largely mutually exclusive expression pattern, with a sharp transition from RUNX1$^+$ NKX3.1$^-$ to RUNX1$^-$ NKX3.1$^+$ cells in the proximal region (*Figure 1A,B*; *Figure 1—figure supplement 1A,B*). These proximal luminal cells had a unique histological profile, with a compact organization, intense nuclear hematoxylin staining, and increased nuclear-to-cytoplasmic ratio (*Figure 1—figure supplement 1C*). In contrast, distal luminal cells had a large cytoplasm with intense pink eosin staining, likely reflecting their secretory function. These observations suggest that RUNX1 marks a subset of proximal luminal cells, distinct from the abundant NKX3.1$^+$ luminal population lining the rest of the prostate epithelium.

The proximal site of the prostate has been proposed to be enriched in cells with stem/progenitor properties (*Goldstein et al., 2008*; *Kwon et al., 2016*; *Tsujimura et al., 2002*; *Yoo et al., 2016*). In order to study the regenerative potential of *Runx1*-expressing cells ex vivo, we took advantage of isoform-specific fluorescent reporter mouse models of *Runx1* (*Draper et al., 2018*; *Sroczynska et al., 2009*). *Runx1* expression is controlled by two promoters, P1 and P2, that respectively drive the expression of the *Runx1c* and the *Runx1b* isoform (*Mevel et al., 2019*). We found that *Runx1* expression in the prostate was exclusively mediated by the proximal P2 promoter, in up to 30% of all epithelial EPCAM$^+$ prostate cells (*Figure 1—figure supplement 2A–C*). Flow-cytometry profiling confirmed the enrichment of P2-*Runx1*:RFP in both basal (EPCAM$^+$ CD49f$^{high}$) and luminal (EPCAM$^+$ CD24$^{high}$) lineages of the proximal compared to the distal prostate (*Figure 1C,D*; *Figure 1—figure supplement 2D*). Mirroring our QBIC spatial analysis (*Figure 1—figure supplement 1B*), P2-*Runx1*:RFP was also detected in a large fraction of the VP epithelium (*Figure 1—figure supplement 2D*).

We therefore used the P2-*Runx1*:RFP mouse line to isolate *Runx1* positive (RFP$^+$) and negative (RFP$^-$) epithelial cells from the basal and luminal compartments of all three prostate lobes and evaluated their regenerative potential in organoid culture assays (*Drost et al., 2016*; *Figure 1E*). The proximal and distal regions of the AP were analyzed separately. In line with previous reports, basal cells were more efficient at forming organoids compared to all luminal fractions (*Drost et al., 2016*, *Kwon et al., 2016*). Importantly, in the luminal fraction, proximal RFP$^+$ luminal cells of the AP consistently displayed higher Organoid Formation Capacity (OFC) than the RFP$^-$ fraction (*Figure 1F*). Luminal RFP$^+$ sorted cells of the DLP and VP also had a greater OFC than RFP$^-$ cells (*Figure 1—figure supplement 3A*). In contrast, no significant differences in OFC were observed between basal enriched subsets and distal luminal RFP$^+$ and RFP$^-$ cells. Brightfield assessment revealed that virtually all organoids had a 'solid' aspect, except for the predominantly 'hollow' organoids derived from proximal RFP$^+$ luminal cells (*Figure 1—figure supplement 3B*). To further characterize their lineage potential, we classified organoids into three types based on the expression of specific lineage markers: unipotent 'basal-like' Keratin 5$^+$ (K5$^+$), unipotent 'luminal-like' Keratin 8$^+$ (K8$^+$), or multipotent K5$^+$ K8$^+$ (*Figure 1G,H*; *Figure 1—figure supplement 3C*). Interestingly, AP proximal luminal RFP$^+$-derived organoids were predominantly small unipotent K8$^+$, while the remainder fraction

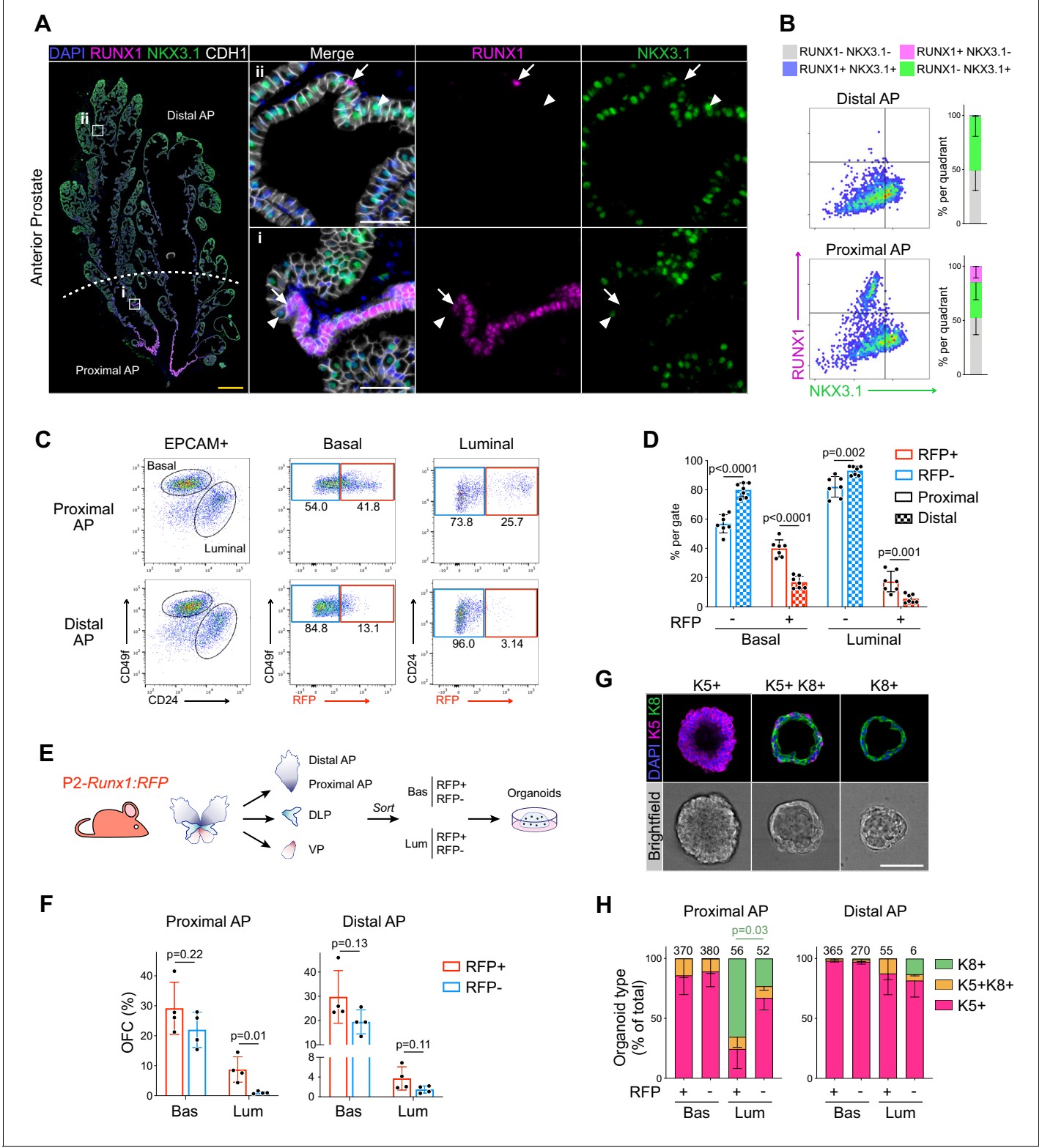

**Figure 1.** RUNX1 marks a subpopulation of mouse proximal prostate luminal cells (PLCs). (**A**) Co-immunostaining of RUNX1, NKX3.1, CDH1 in the mouse Anterior Prostate (AP). Higher magnification images of (**i**) proximal AP and (**ii**) distal AP are shown. Arrows indicate RUNX1+ NKX3.1- cells, arrowheads show RUNX1- NKX3.1+ cells. Scale bars: 500 µm (yellow) and 50 µm (white). (**B**) Quantification of RUNX1 and NKX3.1 nuclear intensity (log10) in CDH1+ epithelial cells by QBIC in proximal and distal AP. n = 6–8 mice. (**C, D**) Flow-cytometry analysis of P2-*Runx1*:RFP mice, and corresponding quantification of the percentages of RFP+ and RFP cells in the basal and luminal fractions of the proximal and distal AP. n = 7 mice. (**E**) Experimental

*Figure 1 continued on next page*

*Figure 1 continued*

strategy to grow organoids from sorted RFP$^+$ and RFP cells from the basal (CD49f$^{high}$) and luminal (CD24$^{high}$) lineages of P2-*Runx1*:RFP mouse reporters. (**F**) Organoid Forming Capacity (OFC) of RFP$^+$ and RFP$^-$ basal and luminal sorted cells after 7 days in culture. *n* = 4 mice. (**G**) Whole-mount immunostaining of unipotent K5$^+$, unipotent K8$^+$ or multipotent K5$^+$ K8$^+$ organoids. Scale bar: 50 µm. (**H**) Quantification of the type of organoids characterized by whole-mount immunostaining, as in G. Numbers of organoids quantified are shown above the graph. p Value is indicated for the proportion of K8$^+$ organoids between Proximal AP Luminal RFP$^+$ versus RFP$^-$ derived subset. Other comparisons were not statistically significant. *n* = 2 mice per group. Source files are available in *Figure 1—source data 1*.

The online version of this article includes the following source data and figure supplement(s) for figure 1:

**Source data 1.** Source data files for *Figure 1*.
**Figure supplement 1.** RUNX1 is enriched in the mouse prostate epithelium.
**Figure supplement 1—source data 1.** Source data files for *Figure 1—figure supplement 1*.
**Figure supplement 2.** *Runx1* expression is mediated by the P2 promoter in the mouse prostate epithelium.
**Figure supplement 2—source data 1.** Source data files for *Figure 1—figure supplement 2*.
**Figure supplement 3.** Characterization of P2-Runx1:RFP derived mouse prostate organoids.
**Figure supplement 3—source data 1.** Source data files for *Figure 1—figure supplement 3*.

mainly gave larger unipotent K5$^+$ organoids (*Figure 1H*; *Figure 1—figure supplement 3D–E*). Few multipotent K5$^+$ K8$^+$ organoids were also identified in nearly all populations.

Together, our results show that RUNX1 marks a specific subset of proximal luminal cells (PLCs), and that its expression in the prostate is mediated by the P2 promoter. RUNX1$^+$ PLCs have a particular predisposition to form unipotent K8$^+$ hollow organoids, suggesting a lineage bias toward the luminal identity, and highlighting differences within the luminal compartment of proximal and distal regions.

### *Runx1*-expressing cells are enriched in the castrated prostate epithelium

In mice, androgen-deprivation can be modeled by surgical castration which leads to prostate regression and enriches for castration-resistant cells (*Toivanen and Shen, 2017*; *Zhang et al., 2018*). This process is accompanied by the death of luminal androgen-dependent cells and a small proportion of basal cells (*English et al., 1987*; *Sugimura et al., 1986b*). To track changes in *Runx1* expression following androgen withdrawal, we surgically castrated P2-*Runx1*:RFP mice and harvested tissue $\geq$4 weeks post-surgery (*Figure 2A*). While intact prostates contained 22.8 ± 6.0% RFP$^+$ epithelial cells, their frequency increased to 87 ± 6.0% following castration (*Figure 2B,C*). High RUNX1 levels were no longer restricted to the proximal region, and RFP was detected in virtually all basal cells of the AP, DLP, and VP, as well as more than 75% of the luminal castration-resistant cells (*Figure 2D*; *Figure 2—figure supplement 1A*). RUNX1-expressing cells often co-expressed TROP2 (*Figure 2E*), known to be widely expressed in castrated prostate epithelium (*Goldstein et al., 2008*; *Wang et al., 2007*). Several castration-resistant luminal populations have been identified in mice (*Barros-Silva et al., 2018*; *Kwon et al., 2016*; *McAuley et al., 2019*; *Tsujimura et al., 2002*; *Wang et al., 2015*; *Wang et al., 2009*; *Yoo et al., 2016*), including rare castration-resistant *Nkx3-1*-expressing cells (CARNs). Accordingly, we observed low, but detectable, levels of NKX3.1 in some luminal cells, but only occasional RUNX1$^+$ NKX3.1$^+$ luminal cells in the distal regions of the castrated prostate (*Figure 2D*; *Figure 2—figure supplement 1B,C*). Importantly, the clear transition from RUNX1$^+$ to NKX3.1$^+$ cells identified in the proximal luminal layer of intact mice was conserved after castration (*Figure 2D,ii*).

Together, these results show that RUNX1 is expressed in the majority of the castration-resistant cells. The RUNX1$^+$ NKX3.1$^-$ subset identified in the proximal luminal epithelium of the intact prostate remain NKX3.1$^-$ following castration, supporting the notion that RUNX1$^+$ PLCs constitute a distinct lineage from distal NKX3.1$^+$ cells.

### scRNA-seq profiling of *Runx1*$^+$ and *Runx1*$^-$ cells in individual lobes of the intact and castrated prostate

To further characterize the RUNX1$^+$ and RUNX1$^-$ fractions residing at different anatomical locations of the prostate, we performed droplet-based single cell (sc)RNA-seq. We sorted EPCAM$^+$ RFP$^+$ and RFP$^-$ cells from individually dissected lobes of intact and castrated prostates isolated from P2-*Runx1*:

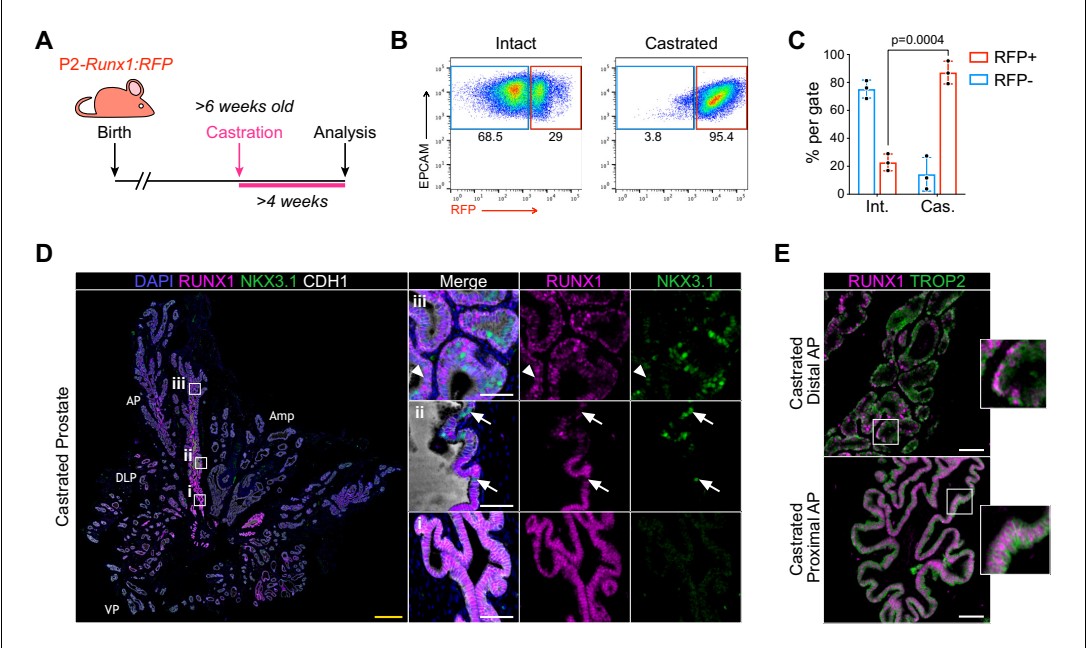

**Figure 2.** RUNX1-expressing cells are enriched in the castrated prostate epithelium. (**A**) P2-*Runx1*:RFP reporter mice were surgically castrated between 6 and 12 weeks of age and analyzed at least 4 weeks post-castration. (**B, C**) Flow-cytometry analysis and corresponding quantification of the proportion of RFP+ and RFP- cells in the EPCAM+ fraction of intact and castrated prostates of P2-*Runx1*:RFP mice. n = 3 mice per group. Int: Intact, Cas: Castrated. (**D**) Co-immunostaining of RUNX1, NKX3.1, CDH1 in the castrated wild-type mouse prostate. Higher magnification images of (**i**) proximal, (**ii**) intermediate, and (**iii**) distal AP are shown. Arrows indicate RUNX1- NKX3.1+ cells, arrowheads show a luminal cell co-stained for RUNX1 and NKX3.1. Amp: ampullary gland. Scale bars: 500 μm (yellow) and 50 μm (white). Int: Intact, Cas: Castrated. (**E**) Co-immunostaining of RUNX1 and TROP2 showing colocalization of the two markers in both proximal (bottom) and distal (top) castrated AP. Scale bars: 50 μm (white). Source files are available in *Figure 2—source data 1*.

The online version of this article includes the following source data and figure supplement(s) for figure 2:

**Source data 1.** Source data files for *Figure 2*.

**Figure supplement 1.** Characterization of RUNX1 expression in the castrated mouse prostate.

**Figure supplement 1—source data 1.** Source data files for *Figure 2—figure supplement 1*.

RFP reporter mice (*Figure 3A*). Sorted populations were multiplexed using MULTI-seq lipid-tagged indices to minimize technical confounders such as doublets and batch effects (*McGinnis et al., 2019b*). We retrieved a total of 3825 prostate epithelial cells from all sorted populations, with a median of 2846 genes per cell (see Materials and methods; *Figure 3—figure supplements 1* and *2A–G*). We identified nine in silico computed clusters expressing canonical epithelial, basal, and luminal markers (*Figure 3—figure supplement 2H–J*). A large population of basal cells was annotated by merging three tightly connected subclusters broadly expressing *Krt5*, *Krt14*, and *Trp63* (*Figure 3B–D*; *Figure 3—figure supplement 2E,J*). Luminal populations expressed surprisingly heterogeneous levels of canonical luminal markers such as *Cd26/Dpp4*, *Cd24a*, *Krt8*, and *Krt18* (*Figure 3—figure supplement 2I*). We annotated those distinct clusters as Luminal-A (Lum-A), Lum-B, Lum-C, Lum-D, Lum-E, and Lum-F (*Figure 3B*). Differential gene expression analysis revealed genes strongly associated with each luminal subpopulation (*Figure 3C and D*; *Figure 3—figure supplement 3A*; *Supplementary file 2*).

Initially, we sought to evaluate the effect of androgen withdrawal on lobe-specific cellular heterogeneity. Lum-A/B/C/D were largely enriched in luminal cells originating from intact prostates, whereas Lum-E/F contained mainly castrated luminal cells (*Figure 3E*; *Figure 3—figure supplement 3B*). Interestingly, Lum-A/C/F mainly contained VP cells, while Lum-B/D/E had a majority of AP and DLP cells, indicating that the lobular identity of luminal cells in the intact prostate is conserved following castration (*Figure 3F*; *Figure 3—figure supplement 3C*). These results suggest that a subset of intact Lum-A/C might undergo partial reprogramming during castration-induced regression and

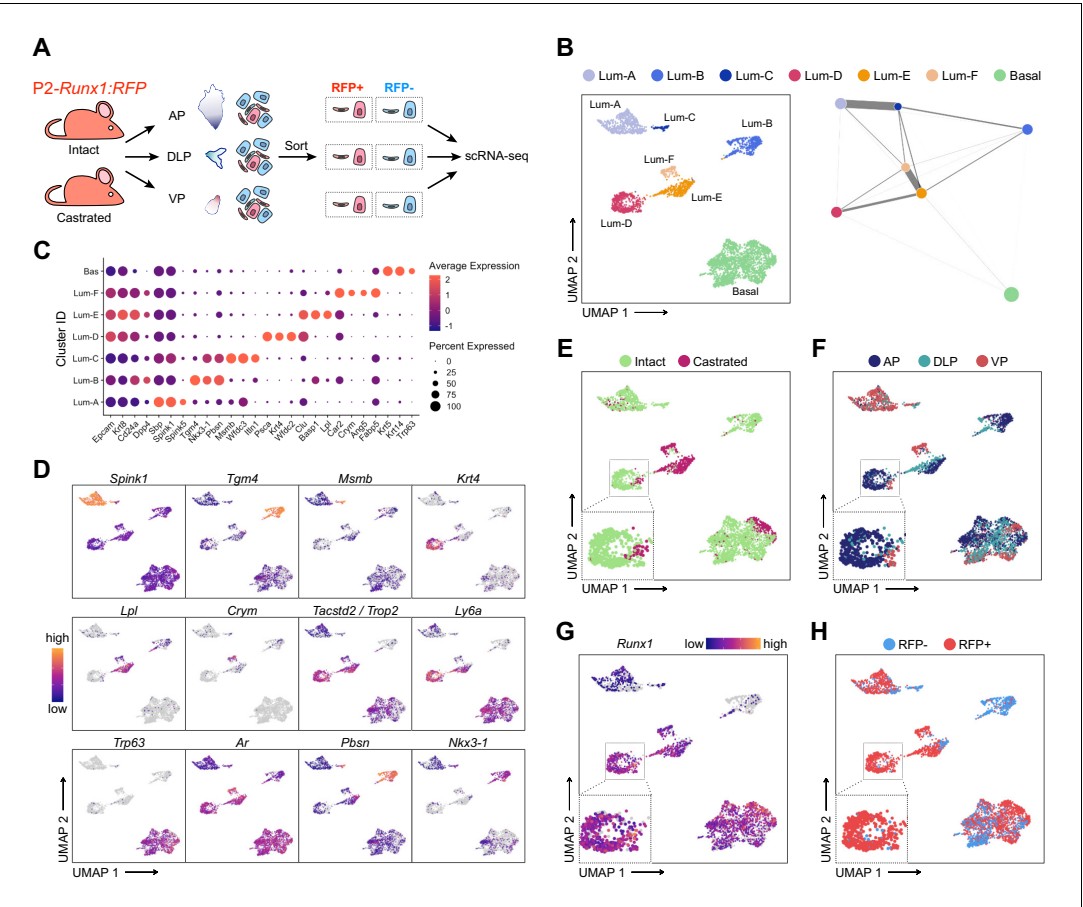

**Figure 3.** scRNA-seq profiling of intact and castrated *Runx1*[+] cells reveals transcriptomic similarity between proximal luminal cells and castration-resistant cells. (**A**) Experimental strategy for scRNA-seq on RFP[+] and RFP[-] cells individually dissected lobes of intact and castrated prostates isolated from P2-*Runx1*:RFP reporter mice. (**B**) UMAP visualization (left) and graph-abstracted representation (PAGA, right) of prostate epithelial cells (*n* = 3,825 cells from three independent experiments). Colors represent different clusters. In PAGA, clusters are linked by weighted edges that represent a statistical measure of connectivity. (**C**) Dot plot showing the expression of selected marker genes associated with each cluster. (**D–H**) UMAP visualization of prostate epithelial cells. Cells in **D** and **G** are colored by a gradient of log-normalized expression levels for each gene indicated. Cell colors in **E** represent the treatment of origin (intact, castrated), in **F** individual lobes of origin (AP, DLP, VP), and in **H** RFP FACS gate of origin (RFP[+], RFP[-]). The online version of this article includes the following figure supplement(s) for figure 3:

**Figure supplement 1.** Pre-processing of the scRNA-seq dataset of adult intact and castrated mouse prostates.

**Figure supplement 2.** Characterization of the scRNA-seq prostate epithelial subset.

**Figure supplement 3.** Characterization of the scRNA-seq prostate epithelial dataset.

**Figure supplement 4.** Gene Ontology and differential expression analysis within the scRNA-seq prostate epithelial dataset.

gives rise to the Lum-F cluster. Similarly, surviving Lum-B/D may predominantly reprogram into Lum-E cells upon castration. Alternatively, the small fraction of intact cells observed in Lum-E and Lum-F clusters might give rise to the expanded Lum-E/F clusters upon castration. In contrast to luminal cells, castrated basal cells were minimally affected by androgen-deprivation and clustered together with intact basal cells (*Figure 3E*). Overall, these results highlight the dramatic changes occurring upon androgen deprivation in the representation of distinct luminal subpopulations.

### *Runx1*-expressing luminal cells are transcriptionally similar to castration-resistant cells

We next specifically focused our attention on RUNX1[+] luminal cells. The Lum-D cluster predominantly consisted of AP-derived RFP[+] cells, as well as a small number of RFP[+] DLP and VP cells (*Figure 3F,H*; *Figure 3—figure supplements 2E*, *3B and C*). High *Runx1* expression in Lum-D

correlated with higher levels of *Tacstd2*/*Trop2*, Ly6 family members as well as *Runx2* (*Figure 3D,G*; *Figure 3—figure supplement 3D,E*). In contrast, *Runx1* was barely detected in clusters Lum-B/C which expressed high levels of *Nkx3-1* while Lum-A cells expressed low levels of both *Runx1* and *Nkx3-1*. These results suggest that the Lum-D cluster corresponds to the distinct RUNX1[+] luminal cells identified in the proximal region of all three prostate lobes (*Figure 1*).

To further characterize the specificities of those populations, we performed gene ontology analysis. In line with the secretory role of distal luminal cells, clusters Lum-A/B/C were enriched in enzymatic activity and protein synthesis functions. In contrast, the Lum-D cluster was enriched in terms related to epithelial developmental processes, similar to Lum-E/F (*Figure 3—figure supplement 4A–I*). This was supported by partition-based graph abstraction (*Wolf et al., 2019*), which uncovered a strong degree of connectivity between the mainly intact Lum-D and castrated Lum-E population (*Figure 3B*). Additionally, the Lum-D cluster contained a small, but defined, subpopulation of castrated epithelial cells, suggesting the preservation of its identity upon androgen deprivation (*Figure 3E,F*). In this population, we found very few genes significantly differentially expressed between intact and castrated cells (n = 103; *Supplementary file 3*). As expected, androgen-regulated genes including *Psca* and *Tspan1* were downregulated in the castrated subset, while strong contributors of the Lum-D identity such as *Tacstd2*/*Trop2*, *Krt4* and *Runx1* did not vary (*Figure 3—figure supplement 4J*). These observations further support the hypothesis that Lum-D/RUNX1[+] PLCs maintain their identity following androgen-deprivation.

Overall, our single-cell transcriptomic analysis highlighted a vast degree of heterogeneity within and between the luminal compartments of both intact and castrated mouse prostates. The tight transcriptional relationship observed between high *Runx1* expressing clusters Lum-D and Lum-E/F suggest that the Lum-D population, which corresponds to PLCs, may contain intrinsically castration-resistant luminal cells.

## Lineage tracing of *Runx1*-expressing cells establishes the intrinsic castration-resistant properties of the proximal luminal lineage

To determine if RUNX1[+] PLCs were enriched in castration-resistant cells, we combined prostate regression-regeneration assays with genetic lineage tracing using *Runx1*[mER-CRE-mER]*Rosa*[Lox-Stop-Lox-tdRFP] mice (*Luche et al., 2007*; *Samokhvalov et al., 2007*), henceforth *Runx1*[CreER] *Rosa26*[LSL-RFP] (*Figure 4A*). Using this model, we could genetically label an average of 4.70 ± 2.8% prostate epithelial *Runx1*-expressing cells with RFP upon tamoxifen injection (*Figure 4B,C*; *Figure 4—figure supplement 1A*). This corresponded to 0.54 ± 0.2‰ of the total epithelium (*Figure 4E*). Consistent with the expression pattern of *Runx1*, the majority of labeled cells were located in the proximal region of the prostate (*Figure 4C*), and co-expressed Keratin 4 (K4) (*Figure 4—figure supplement 1D,E*), previously found enriched in Lum-D cells (*Figure 3D*).

Following surgical castration, we found that the absolute number of RFP[+] marked cells remained stable (*Figure 4—figure supplement 1C,D*). However, the frequency of RFP[+] cells in the epithelial compartment increased by ~4.3 fold (*Figure 4E*; *Figure 4—figure supplement 1B*) indicating that *Runx1*-expressing cells have an enhanced capacity to survive castration compared to *Runx1*-negative cells. Next, we investigated whether these intrinsically castration-resistant *Runx1*-expressing cells were involved in epithelial regeneration upon testosterone addback (*Figure 4B*, bottom). Surprisingly, only 0.71 ± 0.2‰ RFP[+] epithelial cells were found in the regenerated prostate, which was comparable to the intact state (*Figure 4E*; *Figure 4—figure supplement 1B–D*). Although the majority of RFP[+] clones consisted of single cells, we did observe a minor ~2-fold increase in the frequency of larger clones (2–4 cells) after regeneration, highlighting a modest contribution of RFP labeled cells during prostate regeneration (*Figure 4F,G*). We found that most RFP marked cells were luminal K8[+] in intact, castrated, and regenerated prostates (*Figure 4F,H*), with only a few basal K5[+] RFP[+] cells detected in distal areas (*Figure 4F*). Strikingly, more than 90% of all RFP[+] cells remained negative for NKX3.1 in all experimental arms (*Figure 4I*).

Thus, these results indicate that RFP[+] cells, including PLCs, are mostly unaffected by fluctuations in androgen levels during regression-regeneration assays. RUNX1 expression marks intrinsically castration-resistant luminal cells that do not contribute substantially to the expansion of luminal NKX3.1[+] cells during prostate regeneration.

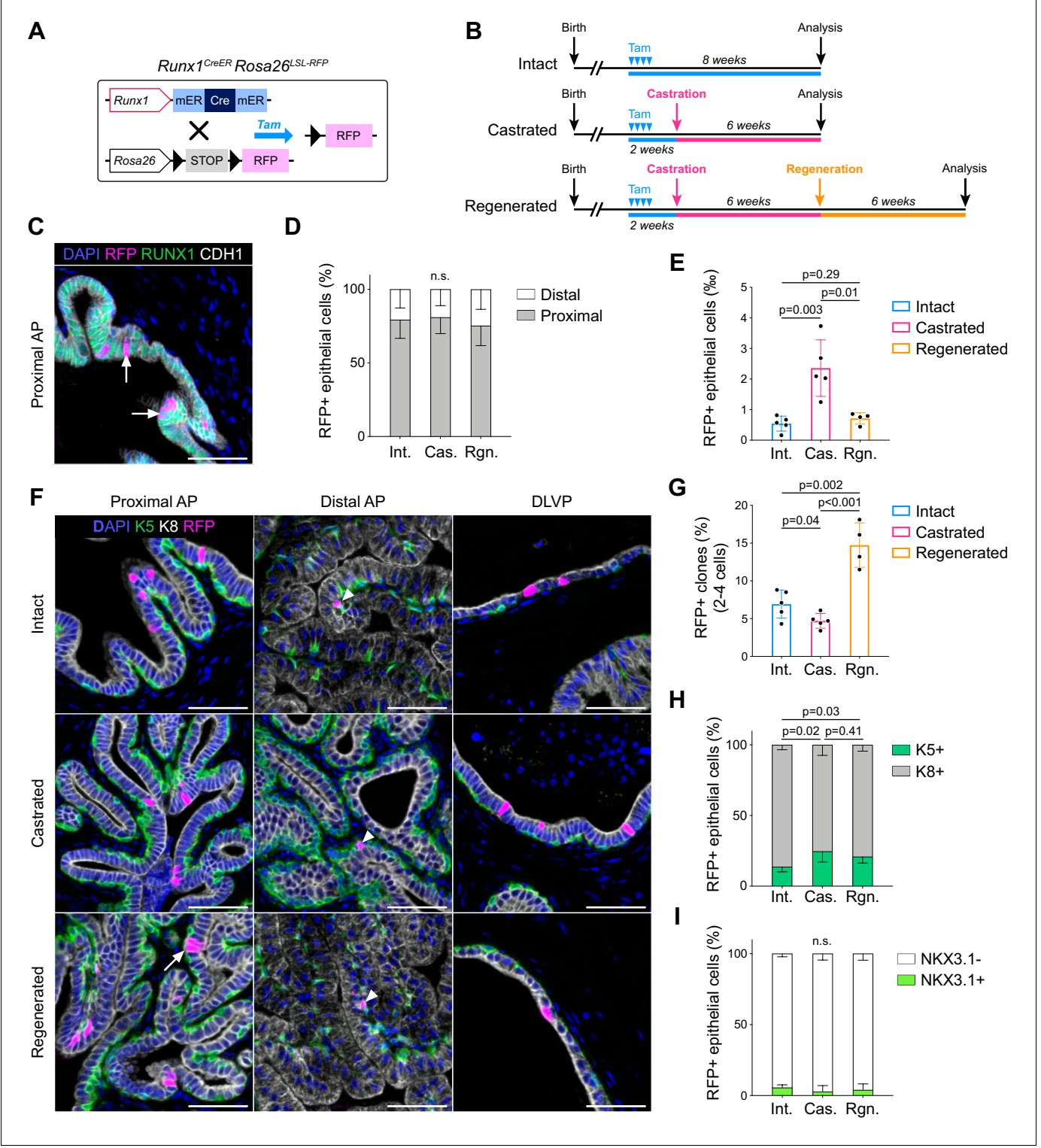

**Figure 4.** Lineage tracing of *Runx1*-expressing cells establishes the intrinsic castration-resistant properties of the proximal luminal lineage. (A) Schematic summary of the genetic lineage-tracing system employed. (B) Experimental strategy for lineage-tracing experiments. (C) Co-immunostaining of RFP, RUNX1, CDH1 in the proximal AP. Arrows indicate RFP labeled RUNX1+ cells. Scale bar: 50 µm. (D) Quantification of the percentage of epithelial RFP+ cells in proximal and distal regions of the prostate in intact (*n* = 5), castrated (*n* = 4) and regenerated (*n* = 4) mice. (E) Quantification of the percentage of epithelial RFP+ cells in intact (*n* = 5), castrated (*n* = 5) and regenerated (*n* = 4) mice. (F) Co-immunostaining of RFP, K5, K8 in the proximal AP, distal AP, and DLVP (DLP + VP). Arrowheads indicate RFP labeled basal cells (K5+) found in distal AP, the white arrow indicates a luminal (K8+) RFP+ clone made of two cells. Scale bar: 50 µm. (G) Quantification of the percentage of epithelial RFP+ clones comprising between two and four

*Figure 4 continued on next page*

eLife Research article

Developmental Biology | Stem Cells and Regenerative Medicine

*Figure 4 continued*

cells in intact (*n* = 5), castrated (*n* = 5) and regenerated (*n* = 4) mice. (**H, I**) Quantification of the percentage of RFP$^+$ cells being K5$^+$ or K8$^+$ in **H**, or NKX3.1$^+$ or NKX3.1$^-$ in **I**, in intact (*n* = 5), castrated (*n* = 5) and regenerated (*n* = 4) mice. Int: Intact, Cas: Castrated, Rgn: Regenerated. Source files are available in *Figure 4—source data 1*.

The online version of this article includes the following source data and figure supplement(s) for figure 4:

**Source data 1.** Source data files for *Figure 4*.
**Figure supplement 1.** Lineage tracing of RUNX1-expressing cells labeled in intact mice.
**Figure supplement 1—source data 1.** Source data files for *Figure 4—figure supplement 1*.

## *Runx1* marks proximal cells during prostate development

Given the singular identity of proximal luminal *Runx1*-expressing cells in the adult prostate, we then asked if this luminal lineage was already emerging during prostate development. At E18.5, once the first prostate buds have emerged, RUNX1 was mainly found in the K8$^{high}$ inner layers of the stratified urogenital epithelium (UGE) (*Figure 5A*). Interestingly, these cells also co-expressed K4 (*Figure 5—figure supplement 1A*), previously found in the Lum D population (*Figure 3D*), as well as LY6D, recently shown to mark a subset of adult luminal progenitors (*Barros-Silva et al., 2018*; *Figure 5—figure supplement 1B*). In contrast, RUNX1 expression was low in p63$^+$ and K5$^+$ cells, either lining the outer UGE or found in the tips of premature NKX3.1$^+$ prostate buds (*Figure 5A–C*). At postnatal day 14 (P14), a prepubescent stage when most of the initial branching events have already occurred (*Sugimura et al., 1986a*; *Tika et al., 2019*), RUNX1 was broadly expressed in the proximal region (*Figure 5D*), mainly in K4$^+$ luminal cells and in some K5$^+$ or p63$^+$ cells (*Figure 5—figure supplement 1C–E*). Conversely, NKX3.1$^+$ cells were found in distal locations, largely distinct from RUNX1$^+$ cells. The specific spatial expression pattern of RUNX1 in proximal luminal cells, largely mutually exclusive with NKX3.1, suggests that these two transcription factors already mark distinct cellular lineages during embryonic prostate organogenesis.

To study the dynamic emergence of RUNX1$^+$ cells during prostate development, we utilized an explant culture system (*Berman et al., 2004*; *Doles et al., 2005*; *Kruithof-de Julio et al., 2013*; *Lopes et al., 1996*). Dissected E15.5 UGS were cultured for up to 7 days in the presence of dihydrotestosterone (*Figure 5E,F*). Bud formation was initiated within 2 days of culture (*Figure 5G*) and composed of a double positive K5$^+$ K8$^+$ stratified epithelium, partially diversifying by day 7 (*Figure 5—figure supplement 2A,B*). On day 0 (E15.5), RUNX1 was detected at the rostral end of the UGE, particularly within the inner layers of the stratified epithelium. After 1 day in culture, NKX3.1 expression emerged in RUNX1$^+$ cells located in the outer layers of the UGE, while defined budding was yet to be observed. On day 2, NKX3.1$^+$ prostate buds were evident and had reduced or absent RUNX1 expression. This pattern was conserved in the mature explant, in which distal tips were mainly NKX3.1$^+$, whereas the proximal area remained RUNX1$^+$ (*Figure 5G,H*), and co-expressed LY6D and K4 (*Figure 5—figure supplement 2C,D*). Cellular proliferation marked by Ki67 was more substantial in distal regions, suggesting that most of the expansion did not take place in the RUNX1$^+$ compartment (*Figure 5—figure supplement 2E*).

These results suggest that prostate budding originates from a subset of cells located in the outer layers of the stratified UGE, transiently marked by RUNX1 and NKX3.1. During embryonic prostate development, *Runx1* expression is already primarily confined to the proximal region of the prostatic ducts, in a distinct compartment from NKX3.1$^+$ cells.

## scRNA-seq of explant cultures reveals the specification of the proximal luminal lineage during embryonic prostate development

The characterization by immunostainings of continuous developmental processes is generally constrained to a small number of markers at a time. To further study the specification of RUNX1 and NKX3.1 lineages, we performed scRNA-seq on UGS explant cultures collected at successive time points: E15.5 (D0), day 1 (D1), day 3 (D3), and day 6 (D6) (*Figure 6A*). After data processing, 3937 developing prostatic cells were retained, with a median of 3608 genes per cell (see Materials and methods; *Figure 6—figure supplement 1*).

Visualization of the dataset using a force-directed layout highlighted the progressive cellular diversification taking place from D0 to D6 (*Figure 6B*). Cellular populations were divided into nine

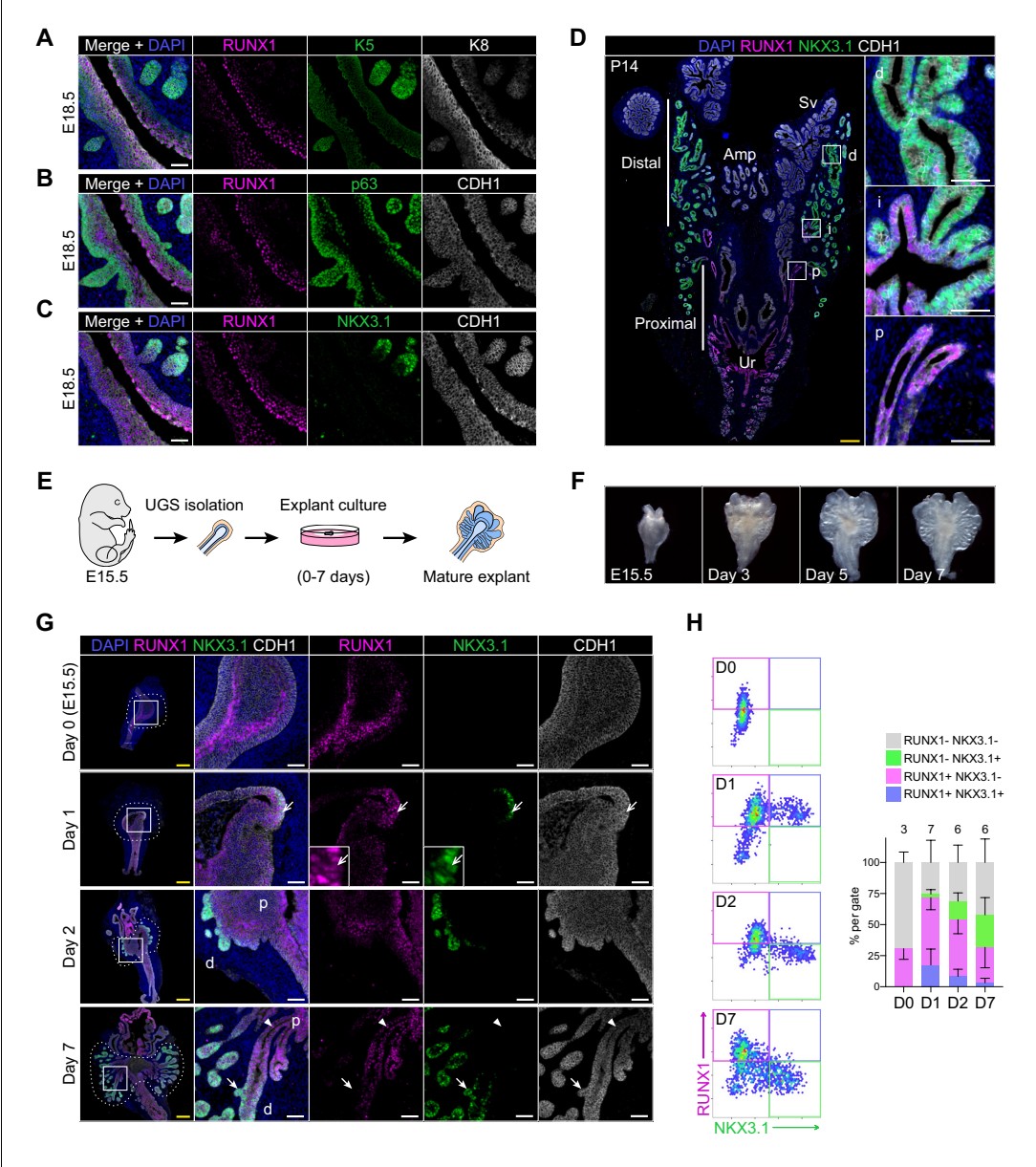

**Figure 5.** RUNX1 marks proximal cells during embryonic prostate development. (**A–C**) Co-immunostainings of the mouse urogenital sinus at E18.5 for RUNX1, K5, K8 in **A**, RUNX1, p63, CDH1 in **B**, RUNX1, NKX3.1, CDH1 in **C**. Scale bar: 50 μm. (**D**) Co-immunostainings of RUNX1, NKX3.1, CDH1 at postnatal (P) day 14. Higher magnification images of (p) proximal, (i) intermediate, and (d) distal regions are shown. Scale bars: 200 μm (yellow) and 50 μm (white). Amp: ampullary gland; Sv: seminal vesicles; Ur: urethra; p: proximal; i: intermediate; d: distal. (**E**) Scheme of the protocol to culture ex vivo explants of mouse UGS harvested at E15.5. (**F**) Representative images of UGS explants at E15.5 (day 0), day 3, day 5, and day 7 of culture showing the formation of premature prostate buds. (**G**) Co-immunostaining of RUNX1, NKX3.1, CDH1 in UGS explants harvested at day 0, day 1, day 2, and day 7. Higher magnification images of each square (left) are shown for each time point. Chevron arrows show RUNX1$^+$ NKX3.1$^+$ cells, closed arrows indicate RUNX1$^-$ NKX3.1$^+$ cells, arrowheads show RUNX1$^+$ NKX3.1$^-$ cells. Scale bars: 200 μm (yellow) and 50 μm (white). (**H**) Quantification of RUNX1 and NKX3.1 nuclear intensity (log$_{10}$) in CDH1$^+$ epithelial cells of UGS explants by QBIC. Quantification was performed within the boundaries delimited in **G** by dotted lines, at day 0 (*n* = 3 explants), day 1 (*n* = 7 explants), day 2 (*n* = 6 explants), and day 7 (*n* = 6 explants). Source files are available in *Figure 5— source data 1*.

The online version of this article includes the following source data and figure supplement(s) for figure 5:

**Source data 1.** Source data files for *Figure 5*.
**Figure supplement 1.** Characterization of RUNX1 expression during prostate development in vivo.
**Figure supplement 2.** Characterization of RUNX1 expression during prostate development in UGS explant cultures.
**Figure supplement 2—source data 1.** Source data files for *Figure 5—figure supplement 2*.

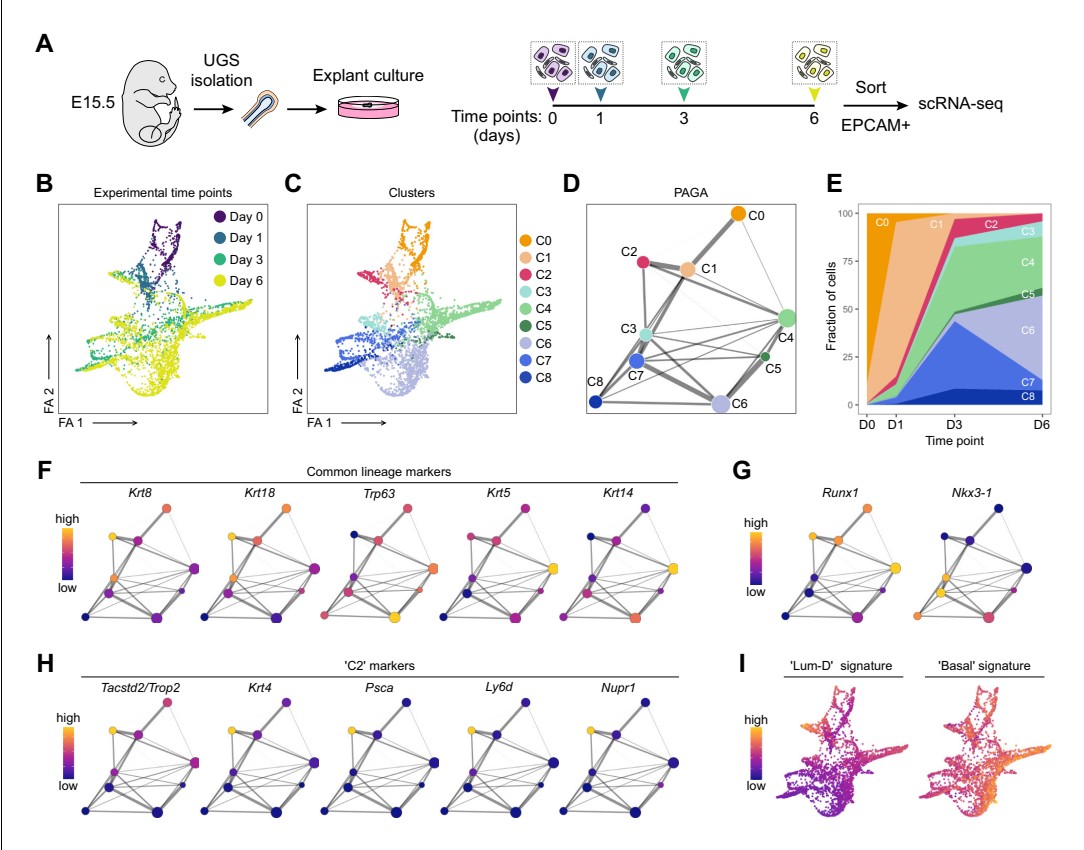

**Figure 6.** scRNA-seq of UGS explant cultures shows specification of the proximal luminal lineage during embryonic prostate development. (A) Experimental strategy for scRNA-seq of UGS explant cultures at day 0, day 1, day 3 and day 6. (B, C) Force directed visualization of the developing prostatic epithelium in UGS explant cultures. In B cells are colored by experimental time points, and in C cells are colored by clusters. (D) PAGA representation of the clusters as in C. Weighted edges between cluster nodes represent a statistical measure of connectivity. (E) Fraction of cells per cluster at each experimental time point, displaying a progressive cellular diversification. (F–H) PAGA representations with cluster nodes colored by a gradient representing the mean log-normalized expression levels of each gene. (I) Force directed visualization of the developing prostatic epithelium in UGS explant cultures. Color gradient represents AUC scores per cell. Per-cell AUC scores were calculated using the 'AUCell' package. Gene signatures for 'Lum-D' (left) and 'Basal' (right) were generated using the list of differentially upregulated genes previously obtained from our adult mouse prostate clusters.

The online version of this article includes the following figure supplement(s) for figure 6:

**Figure supplement 1.** Pre-processing of the scRNA-seq dataset of UGS explant cultures.

**Figure supplement 2.** Characterization of the developing prostatic epithelium in the scRNA-seq dataset of UGS explant culture.

clusters, annotated C0 to C8 (*Figure 6C–E*). C0/C1 contained the majority of D0 and D1 derived cells, while C2-C8 emerged and expanded at later time points. Due to the primitive nature of the UGE at these time points, the classical basal and luminal lineages were not fully established yet (*Figure 6F*; *Figure 6—figure supplement 2A–E*; *Supplementary file 4*). Nevertheless, C4-C6 had a more pronounced 'basal' identity compared to the other clusters. *Krt5/Krt14* marked mainly C4, and additional basal markers including *Trp63, Dcn*, *Apoe,* or *Vcan* were higher in C5/C6. Overall, known regulators of prostate development (*Toivanen and Shen, 2017*) displayed a variable expression pattern across the different clusters. For example, *Foxa1* and *Shh* were strongly expressed in C0/C1, *Notch1* was higher in C3, and *Sox9* in C7 (*Figure 6—figure supplement 2C*), highlighting the potential of this dataset to interrogate specific features of prostate development.

Consistent with our previous results, *Runx1* was highly expressed in clusters having lower *Nkx3-1* levels, including C0, C1, C2, and C4 (*Figure 6G*). To determine how these clusters relate to differentiated prostate lineages, we interrogated population-specific gene signatures previously identified in the adult (*Figure 3*). The 'Basal' signature was enriched across all clusters, especially in C4/C6 (*Figure 6I*; *Figure 6—figure supplement 2F,G*). Strikingly, the 'Lum-D' derived signature was highly

enriched in C2 compared to all the other adult luminal population signatures, suggesting that the 'Lum-D' fate is determined early during prostate development. The singular identity of C2 was characterized by genes previously found highly expressed in the adult 'Lum-D' population, including *Tacstd2/Trop2*, *Krt4*, *Psca*, as well as *Ly6d* and *Nupr1* (*Figure 6H*; *Figure 6—figure supplement 2A*).

Collectively, our scRNA-seq analysis show that adult 'Lum-D'/PLCs share strong similarities with the unique C2 population identified in embryonic explant cultures. This suggests that the distinct proximal luminal lineage is established at the very onset of prostate specification.

## RUNX1+ cells contribute to the establishment of the proximal luminal lineage during embryonic prostate development

To trace the fate of RUNX1+ cells during embryonic prostate specification, we cultured UGS explants isolated from the *Runx1^CreER Rosa26^LSL-RFP* lineage-tracing model. We performed 2 pulses of tamoxifen treatment on day 0 and 1 of culture and analyzed the explants on day 2 and day 7 (*Figure 7A*). The majority of the RFP labeled cells were in the most proximal RUNX1+ subset and rarely found in the distal area of the branches, where RUNX1- cells reside (*Figure 7B,C*). Accordingly, the proportion of RFP+ RUNX1+ cells remained stable between days 2 and 7 (*Figure 7D*). Also, the fraction of RFP+ cells co-expressing p63 remained unchanged throughout the culture (*Figure 7—figure supplement 1A–C*), while a small fraction diversified into either K5+ or K8+ cells (*Figure 7—figure supplement 1D,E*). The scattered RFP+ RUNX1- cells detected in distal branches by day 7 often co-expressed NKX3.1 (*Figure 7E,F*). Overall, this indicates that *Runx1*-expressing cells only marginally contribute to the expansion of the NKX3.1 compartment (*Figure 7G*). Finally, we wondered whether RUNX1+ cells contributed to the establishment of the proximal luminal lineage. We evaluated the proportion of RFP-labeled cells co-expressing K4, previously identified as a marker of the developing C2 and adult Lum-D populations (*Figures 3D* and *6H*). Interestingly, the fraction of K4+ RFP labeled cells increased from 56.9 ± 10.6% to 74.1 ± 3.0% between day 2 and 7 (*Figure 7F,G*). There was also an increase of RFP+ cells expressing *Nupr1*, another marker of the C2 cluster (*Figure 7—figure supplement 1F–H*). Taken together, these results show that only a small subset of *Runx1*-expressing cells contributes to the expansion of NKX3.1+ lineage, found in the distal region of the developing prostatic buds. Instead, the majority of *Runx1*-expressing cells preferentially remain in the proximal region of the premature buds, where the proximal luminal lineage is established.

## Discussion

In this study, we identified RUNX1 as a new marker of a luminal population enriched in the proximal region of the prostatic ducts. By combining scRNA-seq profiling and genetic lineage tracing of *Runx1*-expressing cells, we show that RUNX1+ PLCs present in the intact prostate constitute a developmentally distinct and intrinsically castration-resistant luminal lineage. We propose that proximal and distal lineages are separate luminal entities from the earliest stages of prostate development. As such, our study provides novel insights into the cellular composition and developmental hierarchy of the mouse prostate epithelium.

Until the recent advances in single-cell technologies, the prostate epithelial hierarchy was mainly defined based on anatomical features of the basal and luminal layers, their histological characteristics and the expression of a small subset of markers (*Shen and Abate-Shen, 2010*; *Toivanen and Shen, 2017*). Here, we present two comprehensive scRNA-seq dataset covering both the adult and the developing prostate. To our knowledge, this constitutes the first comprehensive single-cell atlas covering both intact and castrated adult mouse prostates, annotated by their lobe of origin. These datasets can be browsed interactively at http://shiny.cruk.manchester.ac.uk/pscapp/. In particular, our adult scRNA-seq dataset highlighted an extensive degree of cellular heterogeneity, in particular within the luminal epithelia. Several studies recently made similar observations either focusing on the AP (*Karthaus et al., 2020*), the intact prostate (*Crowley et al., 2020*; *Joseph et al., 2020*), or both the intact and castrated prostates (*Guo et al., 2020*). Integration of these multiple datasets will provide a more global view of the transcriptional landscape of the prostate epithelium.

Although mainly known as a master regulator of hematopoiesis, RUNX1 is increasingly implicated in hormone-associated epithelia including malignant conditions such as prostate cancer (*Banach-Petrosky et al., 2007*; *Lie-A-Ling et al., 2020*; *Scheitz et al., 2012*; *Takayama et al., 2015*). Here,

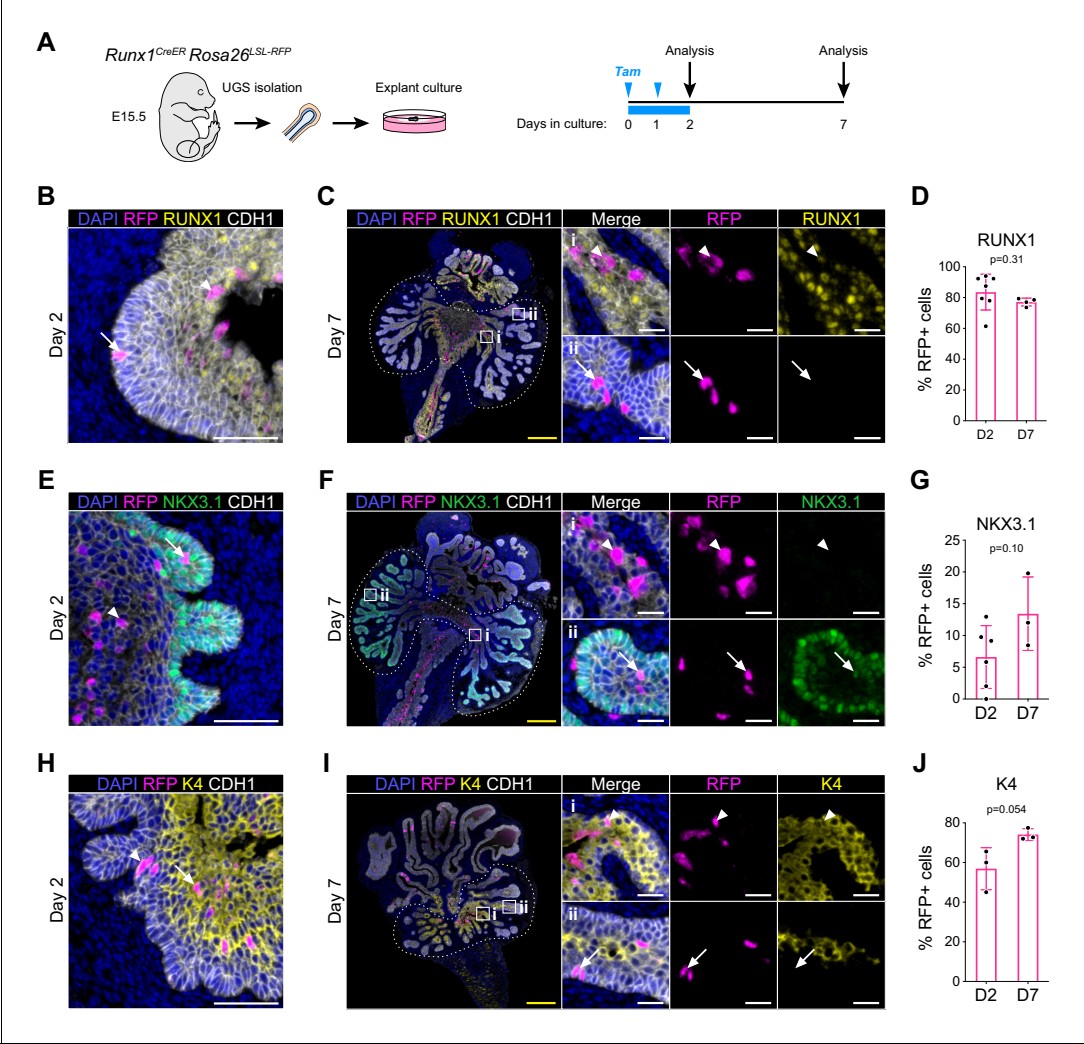

**Figure 7.** RUNX1[+] cells contribute to the establishment of the proximal luminal lineage during embryonic prostate development. (**A**) Strategy for lineage tracing of RUNX1[+] cells in UGS explant cultures. Tamoxifen was applied on day 0 and day 1 and washed out on day 2. (**B, C**) Co-immunostaining of RFP, RUNX1, CDH1 in UGS explants harvested at day 2 (**B**) and day 7 (**C**). Higher magnification images of proximal (**i**) and (**ii**) distal regions are shown for day 7. Arrows show RFP[+] RUNX1-low cells, arrowheads show RFP[+] RUNX1[+] cells. Scale bars: 200 μm (yellow) and 50 μm (white). (**C**) Quantification of the percentage of epithelial RUNX1[+] cells in the RFP subset at day 2 (*n* = 7) and day 7 (*n* = 3) of UGS explant cultures. Quantification was performed within the boundaries delimited in **B** by dotted lines. (**E, F**) Co-immunostaining of RFP, NKX3.1, CDH1 in UGS explants harvested at day 2 (**E**) and day 7 (**F**). Higher magnification images of (**i**) proximal and (**ii**) distal regions are shown for day 7. Arrows show RFP[+] NKX3.1[+] cells, arrowheads show RFP[+] NKX3.1[-] cells. Scale bars: 200 μm (yellow) and 50 μm (white). (**G**) Quantification of the percentage of epithelial NKX3.1[+] cells in the RFP subset at day 2 (*n* = 6) and day 7 (*n* = 4) of UGS explant cultures. Quantification was performed within the boundaries delimited in **F** by dotted lines. (**H, I**) Co-immunostaining of RFP, K4, CDH1 in UGS explants harvested at day 2 (**H**) and day 7 (**I**). Higher magnification images of (**i**) proximal and (**ii**) distal regions are shown for day 7. Arrows show RFP[+] K4 cells, arrowheads show RFP[+] K4[+] cells. Scale bars: 200 μm (yellow) and 50 μm (white). (**J**) Quantification of the percentage of epithelial K4[+] cells in the RFP subset at day 2 (*n* = 3) and day 7 (*n* = 3) of UGS explant cultures. Quantification was performed within the boundaries delimited in **I** by dotted lines. Source files are available in *Figure 7—source data 1*.

The online version of this article includes the following source data and figure supplement(s) for figure 7:

**Source data 1.** Source data files for *Figure 7*.

**Figure supplement 1.** Lineage tracing of RUNX1-expressing cells in UGS explants.

**Figure supplement 1—source data 1.** Source data files for *Figure 7—figure supplement 1*.

we identified a subset of RUNX1[+] luminal cells located in the proximal region of the developing and adult prostate, referred to as RUNX1[+] PLCs, and corresponding to the Lum-D cluster identified in our adult scRNA-seq dataset. Of note, this subset appears to be the equivalent of the 'L2' (*Karthaus et al., 2020*) or 'LumP' (*Crowley et al., 2020*), or 'Lum-C' (*Guo et al., 2020*) clusters

identified in recent studies. In light of the extensive contribution of RUNX transcription factors to developmental processes (*Mevel et al., 2019*), our study suggests that *Runx1*, but also *Runx2*, may be involved in the development and maintenance of specific subpopulations of the prostate epithelium. Future work should therefore aim at characterizing the functional role played by RUNX factors in the prostate, in particular in PLCs.

We demonstrate that these RUNX1[+] PLCs exhibit a greater organoid forming potential compared to the remaining luminal fraction, consistent with previous reports isolating similar proximal populations using different markers such as SCA-1, TROP2 or CD26 (*Crowley et al., 2020*; *Goldstein et al., 2008*; *Guo et al., 2020*; *Karthaus et al., 2020*; *Kwon et al., 2016*). Furthermore, RUNX1[+] PLCs predominantly formed unipotent K8[+] hollow organoids demonstrating their preferential commitment to the luminal fate. The greater clonogenicity of RUNX1[+] PLCs may in fact be linked to the gene expression profile of the corresponding Lum-D population, suggesting a more immature epithelial state, committed to the luminal lineage but not the secretory function of the prostate. Similar to the enhanced regenerative potential of glandular basal cells under specific regenerative conditions (*Centonze et al., 2020*), it is tempting to speculate that these cells act as a latent niche of 'facultative' luminal stem cells (*Clevers and Watt, 2018*), primed to generate a structured prostatic epithelium under defined conditions.

Further characterization of RUNX1 expression in prostate development revealed a consistent expression pattern with the adult. RUNX1[+] luminal cells were restricted to the most proximal region of the developing prostate buds, both in embryos and UGS explant cultures. Our scRNA-seq of the developing prostate revealed a broad basal identity, supporting the presence of multipotent basal progenitors during embryonic development (*Ousset et al., 2012*; *Pignon et al., 2013*), switching to unipotency postnatally (*Tika et al., 2019*). However, we observed a distinct cluster (C2) that strongly resembled the adult Lum-D population, suggesting an early branching event towards the proximal luminal fate at the onset of prostate development. Subsequent lineage-tracing experiments indicated that *Runx1*-expressing cells preferentially populate the emerging proximal luminal identity. It would be interesting to determine if the adult Lum-A, Lum-B, and Lum-C derive from multipotent basal progenitors or from any specific clusters identified in the developing prostate. This appears to be the case at least for the adult Lum-D/RUNX1[+] PLCs which already emerges during embryonic specification.

Our data also sheds a light on the regenerative potential of specific epithelial populations. Basal and luminal lineages have previously been shown to be largely self-sustained using generic basal and luminal *Cre* drivers (*Choi et al., 2012*; *Ousset et al., 2012*). However, whether distinct subpopulations of luminal cells contribute to the regeneration of the others remains poorly understood (*Wang et al., 2009*; *Yoo et al., 2016*). Our characterization of RUNX1[+] PLCs and the detection of a wide variety of luminal populations in our adult prostate scRNA-seq data highlights the possible existence of multiple self-contained luminal populations. Indeed, *Runx1*-driven genetic-tracing experiments in regression-regeneration assays revealed that RUNX1[+] PLCs did not contribute substantially to the regeneration of distal NKX3.1[+] cells. It was however evident that RUNX1[+] PLCs are intrinsically castration resistant and capable of sustaining their own lineage in the regenerated prostate. Recently, it was proposed that prostate epithelial regeneration is driven by almost all luminal cells persisting in castrated prostates (*Karthaus et al., 2020*). Our results are compatible with this model, but we further demonstrate that not all luminal subsets retain the same in vivo regenerative potential in response to androgen stimulation. Thus, we suggest that the model of self-sustained basal and luminal populations might be extended to individual luminal subpopulations. This hypothesis should be tested in the future using a more specific Lum-D *Cre* driver (e.g. *Krt4/Psca*). It will also be of interest to investigate the self-sustenance of other luminal compartments using Lum-A, Lum-B, and Lum-C specific *Cre* drivers.

Finally, our study suggests that the emerging C2/Lum-D population retains a more embryonic-like program, which may relate to their intrinsic castration-resistant potential and have broader relevance to cancer treatment. Along these lines, recent work by Guo and colleagues indicates that *Pten* loss induced in *Psca*-expressing cells of the proximal prostate can initiate prostatic intraepithelial neoplasia (*Guo et al., 2020*). These results warrant future investigation of this luminal subset in the context of cancer development, tumor aggressiveness and treatment responses.

In conclusion, we characterized the expression pattern of *Runx1* in the developing, normal and castrated mouse prostate. We observed that *Runx1* marks proximal luminal cells, which is a distinct

luminal lineage emerging early during prostate specification, displaying intrinsic castration-resistant and self-sustaining properties. Our results therefore reveal strong intrinsic lineage differences within the luminal compartment of the prostate epithelium.

# Materials and methods

**Key resources table**

| Reagent type (species) or resource | Designation | Source or reference | Identifiers | Additional information |
|---|---|---|---|---|
| Strain, strain background (*Mus musculus*, male) | ICR (CD-1) wild-type | Envigo | Hsd:ICR (CD-1) | 7–15 week old males |
| Strain, strain background (*Mus musculus*, male) | P1-*Runx1*:GFP | Georges Lacaud lab | | 7–15 week old males |
| Strain, strain background (*Mus musculus*, male) | P2-*Runx1*:RFP | Georges Lacaud lab | | 7–15 week old males |
| Strain, strain background (*Mus musculus*, male) | *Runx1*$^{mER-CRE-mER}$ *Runx1*$^{CreER}$ *Rosa26*$^{LSL-RFP}$ | RIKEN (Japan) **Samokhvalov et al., 2007** | Runx1-MER-Cre-MER | C57Bl/6J background 7–15 week old males |
| Strain, strain background (*Mus musculus*, male) | *Rosa26*$^{Lox-Stop-Lox-tdRFP}$ *Runx1*$^{CreER}$ *Rosa26*$^{LSL-RFP}$ | European Mouse Mutant Archive **Luche et al., 2007** | B6.Cg-Thy1 Gt(ROSA)26 Sortm1Hjf | C57Bl/6J background 7–15 week old males |
| Antibody | Anti-RUNX1 (rabbit monoclonal) | Cell Signaling | Cat: 8529 RRID:AB_10950225 | IHC/IF (1:100) |
| Antibody | Anti-NKX3.1 (rabbit polyclonal) | Athenaes | Cat: AES-0314 | IHC/IF (1:200) |
| Antibody | Anti-CDH1 (goat polyclonal) | R and D Systems | Cat: AF748 AB_355568 | IHC/IF (1:400) |
| Antibody | Anti-p63 (rabbit monoclonal) | Cell Signaling | Cat: 39692 RRID:AB_2799159 | IHC/IF (1:800) |
| Antibody | Anti-K5 (rabbit monoclonal) | Abcam | Cat: ab52635 RRID:AB_869890 | IHC/IF (1:400) |
| Antibody | Anti-K8 (rabbit monoclonal) | Abcam | Cat: ab53280 RRID:AB_869901 | IHC/IF (1:400) |
| Antibody | Anti-K4 (mouse monoclonal) | Abcam | Cat: Ab9004 RRID:AB_306932 | IHC/IF (1:100) |
| Antibody | Anti-LY6D (rabbit polyclonal) | Proteintech | Cat: 17361–1-AP | IHC/IF (1:100) |
| Antibody | Anti-TROP-2 (goat polyclonal) | R and D Systems | Cat: AF1122 RRID:AB_2205662 | IHC/IF (1:200) |
| Antibody | Anti-BrdU (rat monoclonal) | Abcam | Cat: ab6326 RRID:AB_305426 | IHC/IF (1:400) |
| Antibody | Anti-Ki67 (rabbit monoclonal) | Abcam | Cat: ab15580 RRID:AB_443209 | IHC/IF (1:800) |
| Antibody | Anti-RFP (rabbit polyclonal) | Rockland | Cat: 600-402-379 RRID:AB_828391 | IHC/IF (1:400) |
| Antibody | Anti-RFP (rabbit monoclonal) | MBL | Cat: PM005 RRID:AB_591279 | IF (1:200) |
| Antibody | Anti-GFP (rabbit polyclonal) | MBL | Cat: 598 RRID:AB_591816 | IF (1:200) |
| Antibody | EnVision+/HRP Anti-Rabbit | Dako (Agilent) | Cat: K4003 RRID:AB_2630375 | IHC/IF Ready to use |
| Antibody | EnVision+/HRP Anti-Rabbit | Dako (Agilent) | Cat: K4001 RRID:AB_2827819 | IHC/IF Ready to use |

*Continued on next page*

*Continued*

| Reagent type (species) or resource | Designation | Source or reference | Identifiers | Additional information |
|---|---|---|---|---|
| Antibody | ImmPRESS HRP Anti-Goat | Vector Laboratories | Cat: MP-7405 RRID:AB_2336526 | IHC/IF Ready to use |
| Antibody | ImmPRESS HRP Anti-Rat | Vector Laboratories | Cat: MP-7444 RRID:AB_2336530 | IHC/IF Ready to use |
| Antibody | Donkey anti-Goat IgG 647 | ThermoFischer Scientific | Cat: A-21447 RRID:AB_141844 | IF (1:400) |
| Antibody | Anti-CD16/32 Fc block | Biolegend | Cat: 101301 Clone: 93 RRID:AB_312800 | FACS (1:200) |
| Antibody | Anti-CD45 SB436 | ThermoFischer Scientific | Cat: 62-0451-82 Clone: 30-F11 RRID:AB_2744774 | FACS (1:200) |
| Antibody | Anti-EPCAM BV421 | Biolegend | Cat: 118225 Clone: G8.8 RRID:AB_2563983 | FACS (1:200) |
| Antibody | Anti-EPCAM APC | Biolegend | Cat: 118214 Clone: G8.8 RRID:AB_1134102 | FACS (1:200) |
| Antibody | Anti-CD49f FITC | Biolegend | Cat: 313606 Clone: GoH3 RRID:AB_345300 | FACS (1:200) |
| Antibody | Anti-CD49f APC | Biolegend | Cat: 313616 Clone: GoH3 RRID:AB_1575047 | FACS (1:200) |
| Antibody | Anti-CD24 BV786 | BD Biosciences | Cat: 744470 Clone: M1/69 RRID:AB_2742258 | FACS (1:200) |
| Sequence-based reagent | MULTI-seq reagents | Zev Gartner lab *McGinnis et al., 2019b* | | |
| Software, algorithm | R v3.6.3 | CRAN R Project | SCR_001905 | https://cran.r-project.org |
| Software, algorithm | deMULTIplex | *McGinnis et al., 2019b* | | https://github.com/chris-mcginnis-ucsf/MULTI-seq |
| Software, algorithm | DoubletFinder | *McGinnis et al., 2019a* | SCR_018771 | https://github.com/chris-mcginnis-ucsf/DoubletFinder |
| Software, algorithm | Seurat v3.1.5 | *Satija et al., 2015;* Rahul Satija lab | SCR_016341 | https://github.com/satijalab/seurat |
| Software, algorithm | Scanpy v1.4.6 PAGA | *Wolf et al., 2019* | SCR_018139 | https://scanpy.readthedocs.io/en/stable/ |
| Software, algorithm | AUCell v1.8.0 | *Aibar et al., 2017* | | https://github.com/aertslab/AUCell |
| Software, algorithm | scater v1.14.6 | Bioconductor | SCR_015954 | https://bioconductor.org/packages/release/bioc/html/scater.html |
| Software, algorithm | QuPath v0.2 | *Bankhead et al., 2017* | SCR_018257 | https://qupath.github.io/ |
| Software, algorithm | Cellranger v3.1.0 | 10x Genomics | SCR_017344 | |
| Software, algorithm | FlowJo v10 | BD Life Sciences | SCR_008520 | |
| Software, algorithm | Harmony | PerkinElmer | SCR_018809 | |
| Software, algorithm | Graphpad Prism v8.4.2 | Graphpad | SCR_002798 | |

## Animal work

Animal experiments were approved by the Animal Welfare and Ethics Review Body (AWERB) of the Cancer Research UK Manchester Institute and conducted according to the UK Home Office Project Licence (PPL 70/8580). Genetic lineage-tracing experiments were performed at the Beatson Biological Services Unit (PPL 70/8645 and P5EE22AEE) and approved by the University of Glasgow AWERB. Mice were maintained in purpose-built facility in a 12 hr light/dark cycle with continual access to food and water.

Immunocompetent wild-type ICR (CD-1) mice were purchased from Envigo. P1-*Runx1*:GFP and P2-*Runx1*:RFP have been described previously (*Draper et al., 2018*; *Sroczynska et al., 2009*). Colonies were maintained on a ICR (CD-1) background. C57Bl/6J *Runx1*$^{\text{mER-CRE-mER}}$ (*Samokhvalov et al., 2007*) were provided by RIKEN (Japan). C57Bl/6J *Rosa26*$^{\text{lox-stop-lox-tdRFP}}$ mice (*Luche et al., 2007*) were acquired from the European Mouse Mutant Archive (EMMA). For all transgenic lines, routine genotyping was undertaken at weaning (3 weeks of age) by automated PCR genotyping (Transnetyx). For timed mating experiments, vaginal plug detection was considered as embryonic day (E) 0.5.

All animal procedures were performed on adult males at least 7 weeks of age. Surgical castration was carried out under aseptic conditions. For prostate regeneration assays, testosterone pellets (Belma Technologies) were implanted subcutaneously. For in vivo genetic lineage-tracing experiments, tamoxifen (Sigma, T5648) was resuspended in ethanol and diluted in corn oil at a concentration of 10 mg/mL and administered via intra-peritoneal injections daily for 4 consecutive days using the following regimen: 3 mg, 2 mg, 2 mg, 2 mg.

## Isolation of mouse prostate cells

All dissections were performed under a stereo microscope in sterile PBS. Dissociated murine prostate cells were obtained by digesting pre-minced prostate tissue for 1 hr at 37°C in digestive medium prepared in prepared in ADMEM/F12 (Gibco), and containing 1 mg/mL Collagenase Type I (ThermoFischer Scientific, #17018029), 1 mg/mL Dispase II (ThermoFischer Scientific, #17105041), 10% Fetal Bovine Serum (Gibco), 1% Penicillin-Streptomycin-Glutamine (Sigma), and 10 μM Y-27632 dyhydrochloride (Chemdea, #CD0141). For embryonic urogenital sinuses (UGS), dissociation time was reduced to 30 min. Single cells were obtained after an additional 10 min incubation in TrypLE (Gibco) at 37°C before mechanical dissociation with a syringe and needle (25G). Cells were then filtered through a 50-μm cell strainer.

## Flow-cytometry and cell-sorting

Single cell suspensions were kept in Advanced DMEM/F-12 (Gibco) containing 5% FBS supplemented with 10 μM Y-27632. Cells were incubated for 10 min using unconjugated anti-mouse CD16/32 antibody (Biolegend, C93, #101301) at 4 °C prior to labeling with specific fluorochrome-labeled antibodies. Details of FACS reagents and antibodies are listed in the Key Resources Table. Cells were filtered through a 50 μm cell strainer prior to acquisition. Hoechst 33258 or Sytox blue (ThermoFischer Scientific) were used as viability stains. Single-cell suspensions were analyzed on a Fortessa (BD Biosciences) and sorts were performed on a FACSAriaIII (BD Biosciences). FACS data were analyzed using FlowJo software (BD Life Sciences).

## Organoid formation assays

In vitro organoid formation assays were performed as described in *Drost et al., 2016*. Single cells were resuspended in 40 μL drops of phenol red-free Cultrex RGF BME Type 2 (BME 2, Amsbio, #3533-005-02), and seeded in CellCarrier-96 Ultra Microplates (PerkinElmer, #6055302). Defined organoid culture medium was prepared with Advanced DMEM/F-12 (Gibco), supplemented with 10 mM Hepes (Sigma), Gutamax (Gibco), Penicillin/Streptomycin (Sigma), B27 (Life Technologies, 17504–044), 50 mg/mL EGF (PeproTech, #AF-100–15), 500 ng/mL R-spondin 1 (R and D Systems, #4645-RS), 100 ng/mL Noggin (R and D Systems, #6057 NG), 10 mM Y-27632 dyhydrochloride (Chemdea, #CD0141), 200 nM A83-01 (Tocris Bioscience, #2939), 1.25 mM N-Acetylcystein (Sigma), and 1 nM Dihydrotestosterone (DHT, Sigma #730637). Medium was refreshed every 2–3 days, and organoid cultures were scored after 7 days.

## UGS explant cultures

UGS explant cultures were performed as described previously (*Kruithof-de Julio et al., 2013*). Briefly, E15.5 embryos were obtained from timed matings. Urogenital sinuses (UGS) were isolated from the embryos and cultured using a Durapore Membrane Filter 0.65 µm (#DVPP02500) placed on a stainless-steel mesh for up to 7 days in Ham's F-12/DMEM (Gibco) supplemented with Insulin-Transferrin-Sodium Selenite Supplement (Roche) and 10 µM dihydrotestosterone (Sigma). Media were renewed every 2–3 days. For lineage-tracing experiments, tamoxifen-induced labeling was performed using 0.5 µM 4-hydroxytamoxifen (#T176, Sigma).

## Immunohistochemistry

Prostate tissues were harvested and fixed in 10% buffered formalin for 24 hr. Fixed tissues were processed using standard procedures and embedded in paraffin. Formalin-fixed paraffin-embedded (FFPE) sections (4 µm) were cut and dried overnight at 37°C. Multiplexed immunofluorescent stainings of FFPE sections were performed on an automated Leica BOND RX platform using the Opal multiplexing workflow (PerkinElmer). In brief, sections were dewaxed, and rehydrated, and endogenous peroxidase activity was quenched by 10 min pre-treatment with 3% hydrogen peroxide diluted in TBS-T (Tris-Buffered Saline 0.05% Tween-20). Following on-board heat-induced epitope retrieval with citrate buffer (pH 6.0) for 20 min, sections were incubated for 10 min with 10% Casein (Vector Laboratories) diluted in TBS-T. Each staining cycle included a primary antibody incubation for 30 min, followed by buffer washes, and 30 min incubation with HRP -labeled secondary antibodies (Key Resources Table). After further washes, the Tyramide labeled with a fluorophore (Opal 520, Opal 570 or Opal 650, PerkinElmer) was added for a final 10 min. Subsequent antibody stainings were performed by repeating the same procedure, separated by heat-mediated antibody denaturation using citrate buffer (pH 6.0) for 5 min at 95°C. Nuclei were counterstained with DAPI (Sigma) and slides were sealed using ProLong Gold Antifade Mountant (ThermoFischer Scientific). In situ hybridization (ISH) to detect *Nupr1* (ACD, LS 2.5 Mm-Nupr1 #434818) was done using the Multiplex Fluorescent detection kit (ACD) on the automated Leica BOND RX platform following the manufacturer's instructions. Pre-treatment was done using an EDTA based pH 9.0 epitope retrieval solution for 15 min at 88°C followed by 10 min protease incubation. After ISH, antibody staining was carried out using an anti-RFP antibody for 1 hr detected with EnVision HRP anti-rabbit secondary (Agilent) followed by incubation with Tyramide-conjugated Opal 570 (PerkinElmer) as described above. Anti-CDH1 antibody was applied for 1 hr and detected using an anti-goat Alexa Fluor 647 secondary antibody (ThermoFischer Scientific, #A-21447). Staining of frozen sections was performed as described previously (*Thambyrajah et al., 2016*). The list of antibodies used is available in the Key Resources Table.

## Image acquisition and analysis

Whole-slide images were acquired on an Olympus VS120 slide scanner. Images were analyzed using QuPath v0.2 (*Bankhead et al., 2017*). Briefly, annotations were drawn manually to select areas of interest. Nuclear detection was achieved using the 'cell detection' module on the DAPI channel. A classifier was then trained for each batch of images using the random forest algorithm, to detect the epithelial layers based on either CDH1 or K5/K8 stainings. Single-cell intensity measurements were analyzed using R (3.6.3). For Quantitative Imaged-Based Cytometry (QBIC), single-cell intensity measurements were $\log_{10}$ transformed and plotted using the 'geom_hex' function of the ggplot2 R package. QuPath was used to extract representative high-quality raw images of selected areas from whole slide images using the 'Send region to ImageJ' tool. Images used for publication were processed with ImageJ (NIH Image, Maryland, USA). Confocal images were acquired using a Leica TCS SP8 confocal microscope and LAS X Leica software. Images of whole UGS explant culture were taken using a Leica MZ FLIII microscope.

## Whole-mount immunofluorescent staining of organoids

Whole-mount staining was adapted from *Yokomizo et al., 2012*. Organoids were fixed directly in 96-well plates using 4% paraformaldehyde for 1 hr at 4°C. After three washes of 5 min in PBS, organoids were incubated in PBS-BST, containing PBS, 1% milk, 1% BSA, 10% goat serum (Agilent, #X090710), 0.4% Triton X-100. Pre-conjugated primary antibodies, K5 Alexa Fluor 647 (#ab193895,

Abcam) and K8 Alexa Fluor 488 (#ab192467, Abcam) were diluted at 1/400 in PBS-BST and incubated with the organoids overnight at 4°C on a rocking platform. After three washes of 1 hr in PBS-BST at 4°C, organoids were stained with DAPI at 2 µg/mL diluted in PBS-BST and incubated for another 30 min at 4°C on a rocking platform. Images were acquired on an Opera Phenix High Content Screening System using the 10x air and 20x water lenses. Quantitative analysis was performed using the Harmony software on maximum projection images.

## scRNA-seq sample preparation

A detailed description of the samples, replicates, and the corresponding cellular populations used for each sequencing run is provided in *Supplementary file 1*. For the adult mouse prostate dataset, AP, DLP, and VP lobes were micro dissected and pooled from P2-*Runx1*:RFP reporter mice after dissociation. Single live EPCAM$^+$ cells from RFP$^+$ and RFP$^-$ fractions of each lobes were sorted separately (containing a mix of CD49f$^{high}$ basal and CD24$^{high}$ luminal cells). For the UGS explant culture dataset, the middle regions of the explants were micro dissected to enrich for prostatic branching events and pooled by time point after dissociation. Single live EPCAM+ cells were sorted for each independent time point.

## scRNA-seq sample multiplexing

Individually sorted populations were multiplexed using the MULTI-seq protocol (*McGinnis et al., 2019b*). Reagents were kindly provided by Dr. Zev Gartner. In brief, after sorting, cells were washed once in cold serum- and BSA-free PBS. A lipid-modified DNA oligonucleotide and a specific sample barcode oligonucleotide were then mixed and added to the cells at a final concentration of 200 nM each, and incubated in cold PBS for 5 min. Each individual sample to be multiplexed received an independent sample barcode. Next, a common lipid-modified co-anchor was added at 200 nM to each sample to stabilize the membrane bound barcodes. After an additional 5 min incubation on ice, cells were washed two times with PBS containing 1% FBS 1% BSA in order to quench unbound barcodes. Samples were then pooled together and washed once with PBS 1% FBS 1% BSA. After cell counting, cells were loaded in a Chromium Single Cell 3' GEM Library and Gel Bead Kit v3 (10x Genomics). scRNA-seq library preparation, sequencing and pre-processing.

Gene expression (cDNA) libraries were prepared according to the manufacturer's protocol. MULTI-seq barcode libraries were separated from the cDNA libraries during the first round of size selection, and PCR amplified prior to sequencing according to the MULTI-seq library preparation protocol (*McGinnis et al., 2019b*). For the adult mouse prostate dataset, cDNA libraries of 'run 1' and 'run 2' were sequenced on Illumina NovaSeq 6000 System, and 'run 3' was sequenced on Illumina HiSeq 2500. The UGS mouse prostate explant run was also sequenced on Illumina HiSeq 2500. Sequencing data of cDNA libraries were processed using Cellranger v3.1.0 and mapped onto mm10 mouse reference genome. Pre-processing of the MULTI-seq library fastq files was performed using the 'deMULTIplex' (v1.0.2) R package (https://github.com/chris-mcginnis-ucsf/MULTI-seq) to generate a sample barcode UMI count matrix. Detailed quality control metrics of each sequencing run are provided in *Supplementary file 1*.

## Adult mouse prostate dataset analysis
### Quality control and barcode demultiplexing of individual runs

Each run was pre-processed individually prior data integration. Cellranger outputs were loaded into the R package Seurat (v3.1.5). Cells were kept if they had more than 750 detected genes, less than 7500 UMIs and less than 10% mitochondrial transcripts. Sample barcodes were demultiplexed using the HTODemux function implemented in Seurat. Briefly, a negative binomial distribution was used to estimate the background levels based on *k*-means clustering of the barcode normalized counts. Barcodes with values above the 99% quantile were considered 'positive' for a given sample. Cells positive for more than one barcode were considered as 'doublets'. Doublets and negative cells were excluded for all downstream analyses. Thresholds were empirically adjusted to remove additional cells with possible ambiguous classification (*Supplementary file 1*). Of note, in both 'run 1' and 'run 2', a large number of cells were classified 'negative' due to the failed labeling of 'Bar3' (corresponding to 'Intact DLP RFP+' sample). For these runs, we used DoubletFinder (*McGinnis et al., 2019a*) to remove predicted doublets missed out as a consequence of the failed labeling of 'Bar3'. After

classification, barcodes were represented in UMAP space to confirm the purity of the barcode assignment obtained for each sample (*Figure 3—figure supplement 1A*). We obtained a total of 4499 cells from three independent experiments.

## Integration, low dimensional embedding, and clustering

Data aggregation was performed according to the standard integration procedure implemented in Seurat. In brief, each dataset was log normalized, and 3000 variable features were initially computed using the 'vst' method. For integration, 2000 features and 50 dimensions were used as anchors. Integrated data were scaled and the first 50 principal components (PC) were calculated for downstream analyses. Uniform Manifold Approximation and Projection (UMAP) (*McInnes et al., 2018*) was used for visualization. Graph-based louvain clustering was performed on a shared nearest neighbor graph constructed using 20 nearest neighbors for every cell, and a resolution of 0.4, which gave a reasonable segmentation of the data (*Figure 3—figure supplement 1B,C*). Extensive exploration of each cluster based on known marker genes was then carried out to subset prostate epithelial cells. We found 10 prostate epithelial clusters (*Epcam, Krt8, Cd24a, Spink1, Krt19, Tacstd2, Psca, Krt4, Tgm4, Nkx3-1, Pbsn, Msmb, Piezo2, Trp63, Krt5, Krt14*), 3 clusters of hematopoietic cells (*Vim, Ptprc, Cd74, Itgam, Cd3d*), 1 cluster of endothelial cells (*Pecam1*), 1 cluster of fibroblasts (*Vim, Col1a1*) and 1 cluster of mesonephric derivatives (*Svs2, Pax2*) (*Figure 3—figure supplement 1D, E*).

## Analysis of prostate epithelial populations

The same dimension reduction approach described above was performed on the selected prostate epithelial clusters, using a resolution of 0.3 for graph-based louvain clustering. We annotated one large population of basal cells by merging three subclusters highly expressing *Krt5, Krt14* and *Trp63* as we did not discuss the heterogeneity of the basal compartment in this study (*Figure 3B–D*; *Figure 3—figure supplement 2F,I*). We annotated the different luminal clusters expressing higher levels of *Cd26/Dpp4, Cd24a, Krt8* and *Krt18*, as Lum-A, Lum-B, Lum-C, Lum-D, Lum-E and Lum-F. Several genes specifically marked each cluster, including *Sbp/Spink1* in Lum-A, *Tgm4* in Lum-B, *Msmb* in Lum-C, *Psca/Krt4* in Lum-D, *Basp1/Lpl* in Lum-E, and *Crym* in Lum-F (*Figure 3C,D*; *Figure 3—figure supplement 3A*). Data were then imported in Scanpy (v1.4.6) to infer lineage relationships between cellular populations via partition-based graph abstraction (PAGA) implemented in the tl.paga function (*Wolf et al., 2019*). Briefly, a single-cell neighborhood graph (n_neighbors = 50) was computed using the integrated principal components previously calculated in Seurat. PAGA was generated based on our annotated clusters. The final UMAP representation was generated using PAGA-initialized positions to better preserve the global topology of the data. All final data visualizations were generated in R.

## Differential gene expression analysis and gene ontology

Differential gene expression analyses between clusters were performed using the MAST method (*Finak et al., 2015*) implemented in Seurat within the 'FindAllMarkers' and 'FindMarkers' functions. Testing was limited to genes detected in at least 25% of the tested populations (min.pct = 0.25) and showing at least ±0.25 log fold change difference (logfc.threshold = 0.25). The 'g:GOSt' function of the gprofiler2 R package was used to perform functional enrichment analysis on gene ontology terms (GO:BP, biological processes). Genes showing at least 0.50 log fold change enrichment in the group tested were kept.

## UGS explant cultures dataset

A similar strategy was applied for the analysis of the UGS explant culture dataset, with some alterations described below.

## Quality control and barcode demultiplexing

Cells were kept if they had more than 1000 detected genes, and less than 7.5% mitochondrial transcripts. Barcode classification was performed as above, using the 90% quantile in 'HTODemux' (*Figure 6—figure supplement 1A*). We obtained a total of 5,122 cells that passed quality control from the four time points.

## Low dimensional embedding and clustering

The first 50 principal components and 20 neighbors were used for UMAP visualization. Graph-based clustering was done using a resolution parameter of 0.3. We noticed a strong effect of cell cycle using cell cycles genes defined in *Tirosh et al., 2016*. This was particularly evident using the 'CellCycleScoring' function implemented in Seurat (*Figure 6—figure supplement 1B*). To minimize the impact of cell cycle on downstream analyses, the cell cycle scores were regressed out during data scaling. We identified six main clusters, that we annotated based on the expression of several marker genes (*Figure 6—figure supplement 1C-E*). We identified 2 clusters of developing mesonephric derivatives (*Hoxb7*, *Wfdc2*, *Gata3*, *Sox17*, *Pax2*, *Pax8*, *Lhx1*), 1 cluster of developing bladder urothelium (*Upk3a*, *Foxq1*, *Plaur*, *Krt7*, *Krt20*), 1 cluster of mesenchymal cells (*Vim*, *Col3a1*, *Col1a1*, *Pdgfra*, *Zeb1*) and 1 cluster corresponding to the developing prostatic epithelium (*Epcam*, *Krt8*, *Krt5*, *Krt14*, *Krt15*, *Shh*, *Hoxb13*, *Hoxd13*, *Nkx3-1*). We also identified one cluster largely associated with hypoxia and cellular stress ontologies (*Figure 6—figure supplement 1F*).

## Analysis of the developing prostatic epithelium

The same dimension reduction approach was initially applied on the developing prostatic cluster. After graph-based clustering using a resolution of 0.5, 10 clusters were identified and visualized via UMAP (*Figure 6—figure supplement 1G-J*). We computed diffusion components using 'runDiffusionMap' (ncomponents = 20, k = 20) implemented in the scater (v1.14.6) R package. We found the small cluster C9 to be largely diverging from the remainder fraction in diffusion space, therefore it was excluded for downstream analysis (*Figure 6—figure supplement 1K*). We then imported the data in Scanpy and used the first 10 diffusion components to compute a neighborhood graph (n_neighbors = 20) which was used for PAGA. We finally computed a force-direct layout (ForceAtlas2) using PAGA-initialized positions.

## Analysis of gene set activity

Gene signatures were generated from the list of differentially expressed genes by keeping those showing at least 0.50 log fold change enrichment in each given group. Gene lists were used as custom gene sets (*Supplementary file 5*) in the AUCell (*Aibar et al., 2017*) R package (v1.8.0). Briefly, AUCell uses the Area Under the Curve to evaluate the enrichment of a given gene set in each cell, in a ranking-based manner. It outputs an AUC score for each individual cell, which is used to explore the relative expression of the signature. Per cell AUC scores of each signatures were overlayed on the dimension reduction layout and plotted as boxplots to visualize enrichments across the different cellular subsets.

## Data availability

Raw sequencing files and processed gene expression matrices have been deposited in the NCBI Gene Expression Omnibus under the accession number GSE151944. The processed datasets for both mouse adult prostate and UGS prostate explant cultures can be accessed via a searchable R Shiny application available at http://shiny.cruk.manchester.ac.uk/pscapp/. All codes used to process data and generate figures are available on a public GitHub repository at https://github.com/gla-caud/prostate-scRNAseq (*Mevel, 2020* copy archived at swh:1:dir:c8a38de85e999a595715a4e0a41585fd6b94c44f).

## Statistical analyses

Statistical analyses were performed using Graphpad/Prism (v8.4.2). Data are represented as mean ± SD. Unless otherwise specified in the corresponding figure legend, two-tailed unpaired *t*-tests were used to compare means between two groups. Statistical significance was set at p<0.05. For animal model studies, no statistical method was used to pre-determine the sample size. No randomization or blinding was used for in vivo studies.

## Acknowledgements

We thank the laboratories' members for critical reading of the manuscript, in particular Dr. Michael Lie-a-ling, Dr. Alice Lallo and Dr. Catherine Winchester. We thank the staff at the Histology, Flow

Cytometry, Advanced Imaging, Molecular Biology, and Breeding Unit Core facilities of CRUK Manchester Institute, as well as the CRUK Beatson Biological Services Unit and Flow Cytometry Core facility for technical support. We thank Dr. Zev Gartner for kindly providing the MULTI-seq reagents. We thank Dr. Kirsteen Campbell with assistance on the lineage-tracing experiments. We thank Dr. Berenika Plusa and Dr. Roberto de la Fuente with assistance on timed matings experiments. We thank Professor Hans Jorg Fehling and the European Mouse Mutant Archive for providing the Rosa26^Lox-Stop-Lox-tdRFP mice.

## Additional information

### Funding

| Funder | Grant reference number | Author |
| --- | --- | --- |
| Cancer Research UK | C5759/A20971 | Esther Baena<br>Georges Lacaud |
| Cancer Research UK | C596/A17196 | Karen Blyth |

The funders had no role in study design, data collection and interpretation, or the decision to submit the work for publication.

### Author contributions

Renaud Mevel, Conceptualization, Data curation, Formal analysis, Investigation, Visualization, Methodology, Writing - original draft, Writing - review and editing; Ivana Steiner, Investigation, Writing - review and editing; Susan Mason, Laura CA Galbraith, Rahima Patel, Muhammad ZH Fadlullah, Investigation; Imran Ahmad, Hing Y Leung, Pedro Oliveira, Resources; Karen Blyth, Esther Baena, Conceptualization, Supervision, Funding acquisition, Methodology, Writing - review and editing; Georges Lacaud, Conceptualization, Supervision, Funding acquisition, Methodology, Project administration, Writing - review and editing

### Author ORCIDs

Renaud Mevel https://orcid.org/0000-0002-2742-6576
Ivana Steiner http://orcid.org/0000-0002-2744-1952
Georges Lacaud https://orcid.org/0000-0002-5630-2417

### Ethics

Animal experimentation: Animal experiments were approved by the Animal Welfare and Ethics Review Body (AWERB) of the Cancer Research UK Manchester Institute and conducted according to the UK Home Office Project Licence (PPL 70/8580). Genetic lineage-tracing experiments were performed at the Beatson Biological Services Unit (PPL 70/8645 & P5EE22AEE) and approved by the University of Glasgow AWERB. Mice were maintained in purpose-built facility in a 12-hour light/dark cycle with continual access to food and water. All animal procedures were performed on adult males at least 7 weeks of age. Surgical castration was carried out under aseptic conditions.

### Decision letter and Author response

Decision letter https://doi.org/10.7554/eLife.60225.sa1
Author response https://doi.org/10.7554/eLife.60225.sa2

## Additional files

### Supplementary files

- Supplementary file 1. Quality control metrics and metadata of scRNAseq experiments.
- Supplementary file 2. Genes differentially expressed between adult clusters.
- Supplementary file 3. Genes differentially expressed in intact versus castrated Lum-D cells.
- Supplementary file 4. Genes differentially expressed between UGS explants clusters.

- Supplementary file 5. Gene lists used for the analysis of gene set activity.
- Transparent reporting form

## Data availability

Raw sequencing files and processed gene expression matrices have been deposited in the NCBI Gene Expression Omnibus under the accession number GSE151944. The processed datasets for both mouse adult prostate and UGS prostate explant cultures can be accessed via a searchable R Shiny application available at http://shiny.cruk.manchester.ac.uk/pscapp/. All code used to process data and generate figures is available on a public GitHub repository at https://github.com/glacaud/prostate-scRNAseq. (copy archived at https://archive.softwareheritage.org/swh:1:dir:c8a38de85e999a595715a4e0a41585fd6b94c44f).

The following dataset was generated:

| Author(s) | Year | Dataset title | Dataset URL | Database and Identifier |
|---|---|---|---|---|
| Mevel R, Lacaud G | 2020 | Runx1 marks a luminal castration resistant lineage established at the onset of prostate development | https://www.ncbi.nlm.nih.gov/geo/query/acc.cgi?acc=GSE151944 | NCBI Gene Expression Omnibus, GSE151944 |

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
