## [Decision Letter]

**Acceptance summary:**

The authors elegantly identify a specific subpopulation of proximal luminal cells in the prostate that is intrinsically castration-resistant and self-sustained. The authors show that this cell subpopulation emerges early during prostate development and provide new insights into the lineage relationships of the prostate epithelium.

**Decision letter after peer review:**

Thank you for submitting your article "RUNX1 marks a luminal castration resistant lineage established at the onset of prostate development" for consideration by *eLife*. Your article has been reviewed by three peer reviewers, including Wilbert Zwart as the Reviewing Editor and Reviewer #1, and the evaluation has been overseen by Richard White as the Senior Editor.

The reviewers have discussed the reviews with one another and the Reviewing Editor has drafted this decision to help you prepare a revised submission.

The authors characterized the expression of RUNX1 in developing, normal adult, and castrated mouse prostate. RUNX1 is expressed predominately in a proximal luminal cell population in prostatic ducts, which is distinct during prostate development and demonstrates castration-resistance. This RUNX1 population is largely distinct from NKX3.1+ luminal cells. The RUNX1 proximal luminal cells are capable of forming hollow organoids with some multipotency. Upon castration, RUNX2 expression in the proximal luminal epithelium is increased. These castration-resistant RUNX1 + cells are NKX3.1 negative, further providing evidence that the RUNX1+ cells represent a distinct lineage. This is also observed in the developing mouse prostate; there is early developmental specification of RUNX1+ proximal luminal cells. scRNA-seq reveals substantial heterogeneity in the transcriptome of luminal cells and of prostate cells across the different prostate lobes and this heterogeneity persists following castration. Lineage tracing experiments indicate that castration has little impact on RUNX1 expressing proximal luminal cells. Lineage tracing in developing urogenital sinus explants indicates that RUNX1+ cells predominately contribute to the proximal luminal lineage but also may contribute to the NKX3.1+ lineage. Overall this is a thorough and robust study of RUNX1 expression in the mouse prostate.

Essential revisions:

1) As major point, is that I personally feel this story is a developmental biology story in the organogenesis of the mouse prostate. As such, the prostate cancer connection is basically not present at all. Intrinsically, that is not a bad thing, but I feel the authors should rewrite the study as such, to make it more clear that this is not an oncology study. For example, the castration effects as described here, show outgrown of a RUNX1+ castration resistant subpopulation of healthy prostate epithelial cells, and there is no rationale to believe something comparable would happen in tumors. After repositioning the study towards prostate development and response of normal physiology to castration, I believe the paper is well-positioned without any remaining concerns from my end.

2) The study in its current form is well-presented and I find the data convincing and the conclusions generally well supported. The authors suggest that *Runx1* itself, or the *Runx1*+ luminal cells might have some role in prostate cancer. The study falls short in that regards, and thus feels somewhat incomplete, missing one more important piece of data that will hint to that effect more directly. For instance a *Runx1* knockout mouse that shows some prostate phenotype and especially a prostate cancer delay or something; or) a cell-ablation experiment for the *Runx1*+ cells that would demonstrate again some role for these cells if possible in development of prostate cancer; or) a cancer induction experiments using some oncogene/tumor suppressor gene combination crossed with *Runx1*CreER, demonstrating the origin of castration-resistant cancers in the *Runx1*-luminal cells by looking at early events after induction. I realize that any of these experiments will take another 6 months or more given the mouse work necessary, so this would not be realistic.

If the authors do choose to have a oncology component to their paper, they may include in the Discussion section a couple of sentences indicating the possible connection with prostate cancer, to be further elucidated in the future. If the authors however to want to keep a strong cancer-connection in the story, this would also mean substantial amounts of new data would have to be provided to back this up.

---

## [Author Response]

Essential revisions:1) As major point, is that I personally feel this story is a developmental biology story in the organogenesis of the mouse prostate. As such, the prostate cancer connection is basically not present at all. Intrinsically, that is not a bad thing, but I feel the authors should rewrite the study as such, to make it more clear that this is not an oncology study. For example, the castration effects as described here, show outgrown of a RUNX1+ castration resistant subpopulation of healthy prostate epithelial cells, and there is no rationale to believe something comparable would happen in tumors. After repositioning the study towards prostate development and response of normal physiology to castration, I believe the paper is well-positioned without any remaining concerns from my end.

We agree that our study mostly contributes to improving the understanding of normal prostate development and prostate epithelial plasticity. Beyond castration, as a mean of modelling androgen deprivation therapy, we have not directly investigated the link with prostate cancer, nor used prostate cancer models. As suggested by the reviewer, we have now repositioned our study as a developmental story by re-writing several sections of the text, including the Abstract, Introduction, and Discussion, to remove the possible link with prostate cancer from these sections.

2) The study in its current form is well-presented and I find the data convincing and the conclusions generally well supported. The authors suggest that Runx1 itself, or the Runx1+ luminal cells might have some role in prostate cancer. The study falls short in that regards, and thus feels somewhat incomplete, missing one more important piece of data that will hint to that effect more directly. For instance a Runx1 knockout mouse that shows some prostate phenotype and especially a prostate cancer delay or something; or) a cell-ablation experiment for the Runx1+ cells that would demonstrate again some role for these cells if possible in development of prostate cancer; or) a cancer induction experiments using some oncogene/tumor suppressor gene combination crossed with Runx1CreER, demonstrating the origin of castration-resistant cancers in the Runx1-luminal cells by looking at early events after induction. I realize that any of these experiments will take another 6 months or more given the mouse work necessary, so this would not be realistic.If the authors do choose to have a oncology component to their paper, they may include in the Discussion section a couple of sentences indicating the possible connection with prostate cancer, to be further elucidated in the future. If the authors however to want to keep a strong cancer-connection in the story, this would also mean substantial amounts of new data would have to be provided to back this up.

We would like to thank the reviewer for his feedback, and for highlighting the strengths of our paper. We agree with the reviewer that while we propose that RUNX1^+^ proximal luminal cells may have a role in the context of prostate cancer, and in particular upon androgen deprivation therapy, we have not directly investigated this aspect. As mentioned above, we have now re-written several sections of the text to focus the scope of our study on its developmental aspect. We thank the reviewer for his suggestions to investigate the phenotype of *Runx1* loss in the prostate, as well as to induce an oncogenic event in *Runx1*^+^ cells, using *Runx1*-CreER. We agree that it would be very interesting to carry out these experiments, but that these experiments will take 6 months or more given the mouse work necessary and should be more the subject of future studies and a separate manuscript.

As suggested by the reviewer, we have included a small paragraph at the end of the Discussion to speculate about the implications of our findings to prostate cancer, including a reference to recent work by Guo et al. (Guo et al., 2020), which indicates that tumour initiation by *Pten* loss is possible within *Krt4*+/*Psca*+ proximal luminal cells (which corresponds to RUNX1+ PLCs in our study), and that an equivalent cell type may exist in the human prostate, as well as in human prostate tumours.